# Foundation Model is Efficient Multimodal Multitask Model Selector

**Fanqing Meng**
OpenGVLab, Shanghai AI Laboratory
Shanghai Jiao Tong University
mengfanqing33@gmail.com

**Wenqi Shao** [†]
OpenGVLab, Shanghai AI Laboratory
shaowenqi@pjlab.org.cn

**Zhanglin Peng**
The University of Hong Kong

**Chonghe Jiang**
The Chinese University of Hong Kong

**Kaipeng Zhang**
OpenGVLab, Shanghai AI Laboratory

**Yu Qiao**
OpenGVLab, Shanghai AI Laboratory

**Ping Luo** [†]
The University of Hong Kong
OpenGVLab, Shanghai AI Laboratory
pluo@cs.hku.edu

## Abstract

This paper investigates an under-explored but important problem: given a collection of pre-trained neural networks, predicting their performance on each multi-modal task without fine-tuning them, such as image recognition, referring, captioning, visual question answering, and text question answering. A brute-force approach is to finetune all models on all target datasets, bringing high computational costs. Although recent-advanced approaches employed lightweight metrics to measure models' transferability, they often depend heavily on the prior knowledge of a single task, making them inapplicable in a multi-modal multi-task scenario. To tackle this issue, we propose an efficient multi-task model selector (EMMS), which employs large-scale foundation models to transform diverse label formats such as categories, texts, and bounding boxes of different downstream tasks into a unified noisy label embedding. EMMS can estimate a model's transferability through a simple weighted linear regression, which can be efficiently solved by an alternating minimization algorithm with a convergence guarantee. Extensive experiments on 5 downstream tasks with 24 datasets show that EMMS is fast, effective, and generic enough to assess the transferability of pre-trained models, making it the first model selection method in the multi-task scenario. For instance, compared with the state-of-the-art method LogME enhanced by our label embeddings, EMMS achieves 9.0%, 26.3%, 20.1%, 54.8%, 12.2% performance gain on image recognition, referring, captioning, visual question answering, and text question answering, while bringing $5.13\times$, $6.29\times$, $3.59\times$, $6.19\times$, and $5.66\times$ speedup in wall-clock time, respectively. The code is available at https://github.com/OpenGVLab/Multitask-Model-Selector.

---

[†] Corresponding Authors: shaowenqi@pjlab.org.cn; pluo@cs.hku.edu

37th Conference on Neural Information Processing Systems (NeurIPS 2023).

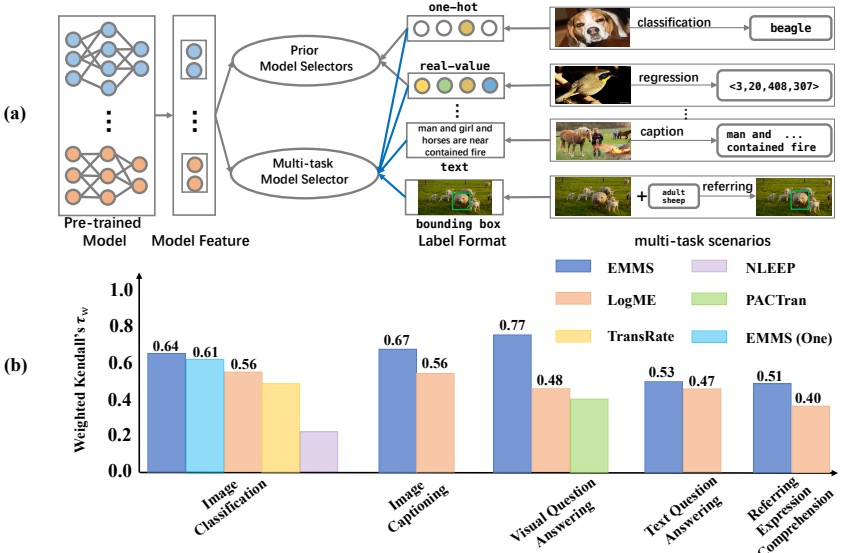

Figure 1: Comparison between prior pre-trained model selectors and our multi-task model selector. (a) denotes that a model selector measures transferability by modeling the compatibility between the model feature and task label. Previous model selectors can only receive labels with one-hot or real-valued vectors. Our multi-task model selector can be employed in various tasks with diverse label formats. (b) denotes that our proposed EMMS is applicable and effective in various downstream tasks while previous transferability metrics can be only used in classification or regression tasks.

## 1 Introduction

Pre-trained models (such as neural network backbones) are crucial and are capable of being fine-tuned to solve many downstream tasks such as image classification [1], image captioning [2], question answering [3], and referring segmentation [4]. This "pre-training → fine-tuning" paradigm shows that the models pre-trained on various datasets (*e.g.,* ImageNet [1]and YFCC100M [5]) by many objectives (*e.g.,* supervised and self-supervised) can provide generic-purpose representation, which is transferable to different tasks. A large number of pre-trained models have been produced with the rapid development of network architecture research, such as convolutional neural networks (CNNs) [6, 7, 8] and transformers [9, 10, 11]. When given a large collection of pre-trained models to solve multiple multi-modal tasks, an open question arises: *how to efficiently predict these models' performance on multiple tasks without fine-tuning them?*

Existing works [12, 13, 14, 15, 16, 17, 18, 19] answered the above question using model selection approaches, which are of great benefit to transfer learning. For example, when a neural network is properly initialized with a better pre-trained checkpoint, it will achieve faster convergence and better performance on a target task [20, 21]. However, it is challenging to quickly identify an optimal one from a large collection of pre-trained models when solving each multi-modal task. This is because of two reasons. Firstly, the ground truth of model ranking can only be obtained by brute-force fine-tuning and hyper-parameter grid search, which are computationally expensive [22]. Secondly, the recent methods [12, 13, 14, 23] that can estimate the transferability of pre-trained models are not generic enough for a variety of multi-modal tasks. For instance, an approach [14, 12] that relies on the prior knowledge of a single specific task would be ineffective in others.

To address the above challenges, we need a unified representation to represent diverse label formats in each multi-modal task *e.g.,* categories, texts and bounding boxes. Existing methods cannot be employed in multi-task scenarios because they only receive labels with one-hot or real-valued vectors, as shown in Fig.1 . For example, LEEP [12] and PACTran [24] are carefully designed for classification tasks. GBC [25] and TransRate [23] measure transferability using class separability, which relies on prior knowledge in classification task. Although LogME [13] can be used in both classification and regression tasks, it relies on real-valued labels, making it inapplicable in other label formats such as text descriptions.

In contrast to the above works, we propose an Efficient Multi-task Model Selector with an acronym, EMMS, which can select the most appropriate pre-trained model for solving each multi-modal task.

This is achieved by employing foundation models, such as CLIP [26] and GPT-2 [27], to transform diverse label formats into a unified label embedding space. The estimated label embedding contains more rich information than the conventional one-hot and real-valued label encoding.

In this way, EMMS can measure the compatibility between the models' features and corresponding label embeddings on various tasks, as shown in Fig. 1 and Fig. 2 . This results in a more generic assessment of the models' transferability than previous methods. Specifically, EMMS treats the estimated label embeddings as noisy oracles of the ground-truth labels, and it turns a log-likelihood maximization problem into a simple weighted linear square regression (WLSR). We propose an alternating minimization algorithm to solve WLSR, which can be solved with a theoretical convergence guarantee efficiently. Extensive experiments validate the effectiveness of EMMS on multiple tasks, including image classification [1], image captioning [2], question answering on both image [28] and text [29], referring comprehension [4], and landmark detection [30].

The **contributions** of this work are summarized as follows. (1) We propose a generic transferability estimation technique, namely Efficient Multi-task Model Selector (EMMS). Equipped with a unified label embedding provided by foundation models and a simple weighted linear square regression (WLSR), EMMS can be fast, effective, and generic enough to assess the transferability of pre-trained models in various tasks. (2) We propose a novel alternating minimization algorithm to solve WLSR efficiently with theoretical analysis. (3) Extensive experiments on 5 downstream tasks with 24 datasets demonstrate the effectiveness of EMMS. Specifically, EMMS achieves 9.0%, 26.3%, 20.1%, 54.8%, 12.2%, performance gain on image recognition, referring, captioning, visual question answering, and text question answering, while bringing $5.13\times$, $6.29\times$, $3.59\times$, $6.19\times$, and $5.66\times$ speedup in wall-clock time compared with the state-of-the-art method LogME enhanced by our label embeddings, respectively.

## 2   Related Work

**Transferability Estimation.** Model selection is an important task in transfer learning. To perform model selection efficiently, methods based on designing transferability metrics have been extensively investigated. LEEP [12] pioneers to evaluate the transferability of source models by empirically estimating the joint distribution of pseudo-source labels and the target labels. But it can only handle classification tasks with supervised pre-trained models because the modeling of LEEP relies on the classifier of source models. Recent works propose several improvements over LEEP to overcome the limitation. For example, NLEEP [31] replaces pseudo-source labels with clustering indexes. Moreover, LogME [13], TransRate [23], and PACTran [24] directly measure the compatibility between model features and task labels. Although fast, these metrics can only be used on limited tasks such as classification and regression. This work deals with model selection in multi-task scenarios. We propose EMMS to evaluate the transferability of pre-trained models on various tasks.

**Label Embedding.** Label embedding represents a feature vector of task labels, which can be generated in various ways. The classical approach is to use one-hot encoding to represent the labels as sparse vectors, which is widely used in image classification. Another way is to transform labels into vectors by embedding layers. For example, an RNN module is employed to generate label representation in [32], which is encouraged to be compatible with input data vectors in text classification tasks. In addition, it is also common to treat the labels as words and use techniques such as word2vec [33] or GloVe [34] to learn vector representations of the labels. The main obstacle in the multi-task scenario is how to deal with diverse label formats. In this work, we follow the idea of word embedding and treat task labels as texts, which are then transformed into embeddings by publicly available foundation models  [26, 27].

**Foundation Models.** CLIP [26] is the first known foundation model which learns good semantic matching between image and text. The text encoder of CLIP can perform zero-shot label prediction because it encodes rich text concepts of various image objects. By tokenizing multi-modal inputs into homogeneous tokens, recent work on foundation models such as OFA [35] and Uni-Perceiver [36] use a single encoder to learn multi-modal representations. In this work, we utilize the great capacity of foundation models in representing image-text concepts to generate label embedding. It is noteworthy that although foundation models can achieve good performance in various downstream tasks, they may not achieve good zero-shot performance on many tasks[37, 38, 39] and it is still computationally expensive to transfer a large model to the target task [40, 41]. On the contrary, a multi-task model

selector can quickly select an optimal moderate-size pre-trained model that can generalize well in target tasks. In this sense, a multi-task model selector is complementary to foundation models.

## 3  Preliminary of Model Selection

**Problem Setup.** A target dataset with $N$ labeled samples denoted as $\mathcal{T} = \{(x^n, y^n)\}_{n=1}^N$ and $M$ pre-trained models $\{\phi_m = (\theta_m, h_m)\}_{m=1}^M$ are given. Each model $\phi_m$ consists of a feature extractor $\theta_m$ producing a $D$-dimension feature (i.e. $\hat{x} = \theta_m(x) \in \mathbb{R}^D$) and a task head $h_m$ outputting predicted label given input $x$ [6, 9]. In multi-task scenarios, the ground-truth label comes in various forms, such as category, caption, and bounding box, as shown in 1 . The task of pre-trained model selection is to generate a score for each pre-trained model thereby the best model can be identified to achieve good performance for various downstream tasks.

**Ground Truth.** The ground truth is obtained by fine-tuning all pre-trained models with hyper-parameters sweep on the target training dataset and recording the highest scores of evaluation metrics [31, 13] (e.g. test accuracy and BLEU4 [42]) . We denote ine-tuning scores of different models as $\{G_m\}_{m=1}^M$. Since fine-tuning all models on all target tasks requires massive computation cost, research approaches design lightweight transferability metrics which offer an accurate estimate of how well a pre-trained model will transfer to the target tasks.

**Transferability Metric.** For each pre-trained model $\phi_m$, a transferability metric outputs a scalar score $T_m$ based on the log-likelihood, as written by

$$T_m = \sum_{n=1}^N \log p(y_n | x_n; \theta_m, h_m) \tag{1}$$

where $(x_n, y_n)$ denotes the $n$-th data point in target dataset $\mathcal{T}$. A higher log-likelihood value for $T_m$ indicates that the model $\phi_m$ is likely to achieve better performance on the intended task. Numerous transferability metrics have been proposed by modeling prediction probability $p(y_n | x_n; \theta_m, h_m)$ in various ways. Although being efficient, they can hardly be used in multi-task scenarios.

**Challenges in Multi-task Scenarios.** Existing transferability metrics fail to generalize to various tasks for two reasons. Firstly, existing methods such as LEEP and LogME can only deal with real-value label formats. But $y_n$ can be a sentence of words in the task of image caption. Secondly, a large number of the previous metrics estimate transferability through the target task's prior information such as maximizing inter-class separability, which is inapplicable in multi-task scenarios except for the classification. To overcome these difficulties, we introduce a simple regression framework with unified label embeddings provided by several foundation models in Sec.4.

## 4  Our Method

In this section, we introduce our Efficient Multi-task Model Selector (EMMS). To overcome the difficulty of diverse label formats, EMMS employs foundation models to transform various labels into unified label embeddings in Sec.4.1. By treating label embeddings provided by multiple foundation models as noisy oracles of ground truth labels, EMMS can calculate transferability metric under a simple weighted linear square regression (WLSR) framework in Sec.4.2. We design an alternating minimization algorithm to solve WLSR efficiently in Sec. 4.3. The illustration of our EMMS is provided in Fig. 2 .

### 4.1  Foundation Models Unify Label Embedding

In general, label embeddings or label representations should encode the semantic information such that two labels with low semantic similarity have a low chance to be grouped. A common scheme is to represent label embedding as a one-hot vector. However, one-hot representation can not embed labels with text formats such as captions in the image caption task. Following the design in multi-modality foundation models [42], we treat labels with diverse formats as a text sequence, which can be encoded by pre-trained foundation models, as shown in Fig. 2 .

**Label Embedding via Foundation Models (F-Label).** Thanks to the great representational capacity, the foundation model can construct label embedding (termed F-label) while preserving its rich

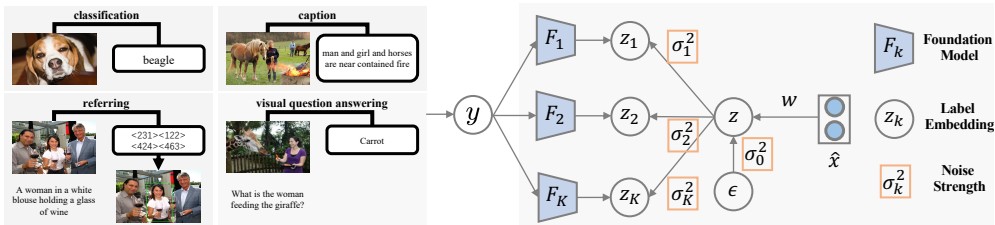

(a) Examples of text labels in various tasks

(b) Graph model of regression with multiple noisy labels

Figure 2: Overview of our EMMS. (a) shows that labels in various tasks can be expressed by texts. (b) presents the graph model of regression with multiple noisy labels. We use several foundation models to encode text labels as label embeddings which are deemed as noisy oracles of true label embedding $z$. Moreover, $z$ is a linear mapping of model feature $\hat{x}$ with Gaussian noise $\epsilon \sim N(0, \sigma_0^2)$.

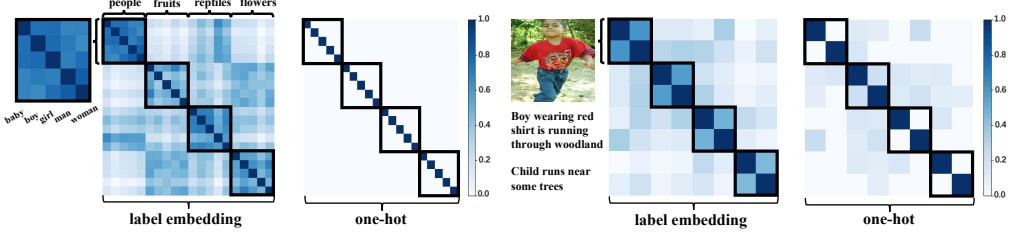

(a) Example of label embedding in image classification

(b) Example of label embedding in image caption

Figure 3: Label embedding has richer semantic information than one-hot labels. (a) indicates that in the classification task, F-Label can capture the correlation of labels with different granularity than one-hot encoding. (b) shows that in the image caption task, F-label can model the semantic relevance of two captions corresponding to the same image better than the one-hot label.

semantic information of labels. Given a label $y$ in the target task, the label embedding $z \in \mathbb{R}^L$ is obtained by $z = F(y)/\|F(y)\|_2$ where $F$ can be instantiated by various foundation models to process diverse label formats. $\ell_2$ normalization is utilized to normalize the representations extracted from different foundation models. Moreover $F$ can be implemented as CLIP [26], BERT [43] and GPT-2 [27] when task label $y$ is text. Note that label embedding extraction can be fast enough with GPU parallel computation. We provide the runtime analysis in Appendix Sec.C.

**Benefits of F-Label.** F-Label has several advantages over one-hot label representations. Firstly, it embeds richer semantic information than one-hot label, leading to accurate modeling of the semantic relationships between different labels. As shown in Fig.3 , F-Label leads to a higher correlation between fine-grained classes than one-hot encoding. Secondly, compared with one-hot labels, F-label can be obtained in a variety of tasks as long as the task label can be transformed into a text sequence. With the assistance of F-Labels, model selection can be established in multi-task scenarios.

## 4.2 Regression with Unified Noisy Label Embeddings

To estimate the transferability of pre-trained models, the relationship between model features $\hat{x} \in \mathbb{R}^D$ and F-Label $z \in \mathbb{R}^L$ should be modeled in order to calculate the transferability score $T_m$ in Eqn. (1). On the other hand, since a semantic label $y$ can be embedded by several foundation models, the label embedding set can be constructed as $\mathcal{Z} = \{z_k = F_k(y)/\|F_k(y)\|_2, k \in [K]\}$ where $\{F_k\}_{k=1}^K$ denotes $K$ foundation models. Now, we utilize data points $\{(\hat{x}_k^n, z_1^n, \cdots, z_K^n)\}_{n=1}^N$ to model the relationship between model features and F-Labels.

**Setup.** As shown in Fig.2 , we assume that true label embedding $z$ is a linear mapping of the model feature with additive Gaussian noise with a variance of $\sigma_0^2$, as given by $z = z_0 + \epsilon = w^T\hat{x} + \epsilon$ and $\epsilon \sim N(0, \sigma_0^2 I_L)$ where $z_0 = w^T\hat{x}$ is the regression prediction, $w \in \mathbb{R}^{D \times L}$ and $\epsilon$ are regression weights and regression error, respectively, and $I_L$ is a L-by-L identity matrix.

We assume that F-labels $\{z_k\}_{k=1}^K$ obtained from different foundation models are oracles that independently provide noisy estimates of the true label embedding $z$. Formally, we have $P(z_k|z) = N(z, \sigma_k^2 I_L)$. By the above setup, EMMS would be performed with noisy labels. Hence, EMMS tends to select pre-trained models robust to the label noise.

**Reasonableness of the Linear Assumption.** Specifically, EMMS assumes that the true label embedding $z$ is a linear mapping of the model feature with Gaussian noise. The linear assumption is reasonable in image and text classification tasks because a linear classifier is usually used when the pre-trained model is transferred to a target task, which is commonly used in recent methods. For example, LogME [13] assumes that: $z \leftarrow N(w^T \hat{x}, \beta^{-1})$, which implies that there is a linear mapping from the model feature space to the label space. PACTran [24] also has a similar setting. The difference is that LogME takes a one-hot label as the true label embedding, which limits its applicability. But our EMMS treat the true label embedding $z$ as an implicit variable. And F-Labels $\{z_k\}_{k=1}^K$ obtained from different foundation models are assumed to be noisy oracles of true label embedding $z$. Since labels in many tasks can be easily encoded into F-Lablels, our EMMS can be used as a multitask model selector. We verify effectiveness the linear assumption in various multi-model tasks with extensive experiments in Sec.5.

**Computation of Log-Likelihood.** To model the relationship between model features and F-Labels, we need to estimate regression weights $w$, strengths of label noises $\{\sigma_k\}_{k=0}^K$. For simplicity of notation, we consider the case $L = 1$, i.e. F-labels are scalars. Given $N$ data points, the log-likelihood is given by

$$\mathcal{L} = N \log A_1 - \frac{N}{2} \log A_2 + \sum_{n=1}^N \left( \frac{(A_3^n)^2}{4A_2} - A_4^n \right) + \text{const} \tag{2}$$

where $A_1 = \prod_{k=0}^K 1/\sigma_k, A_2 = \sum_{k=0}^K 1/2\sigma_k^2, A_3^n = \sum_{k=0}^K z_k^n/\sigma_k^2$, and $A_4^n = \sum_{k=0}^K (z_k^n)^2/\sigma_k^2$. The detailed derivation of Eqn.(2) is provided in the Appendix Sec.A.

**Maximizing Log-likelihood as Weighted Linear Square Regression (WLSR).** The remaining issue is to determine parameters $w$ and $\{\sigma_k\}_{k=0}^K$ by maximizing the log-likelihood in Eqn. (2). But it can be intractable because $w$ and $\{\sigma_k\}_{k=0}^K$ are heavily coupled. To mitigate this issue, we turn the log-likelihood maximization into a weighted linear square regression by rearranging Eqn. (2) as $-\mathcal{L} = \frac{1}{2}\|Xw - Zt\|_2^2 + R(\{\sigma_k\}_{k=0}^K)$, where $X \in \mathbb{R}^{N \times D}$ is the data matrix whose $n$-th row is model feature $(\hat{x}^n)^T$, $w \in \mathbb{R}^{D \times 1}$ are weight parameters, $Z \in \mathbb{R}^{N \times K}$ is F-Label matrix whose $k$-th column is the label embedding $z_k$, and $t \in \mathbb{R}^{K \times 1}$ satisfies that $1_K^T t = 1, t \geq 0$ which is a $(K-1)$-D simplex denoted as $\triangle^{K-1}$. $R(\cdot)$ is a regularization term parameterized with $\{\sigma_k\}_{k=0}^K$. We provide the derivations in Appendix Sec.A.

We note that the computational intractability comes from the data-dependent regularizer $R(\cdot)$. For efficient computation, we drop $R(\cdot)$, turning the log-likelihood maximization into a problem of WLSR, as given by

$$\min_{w \in \mathbb{R}^{D \times 1}, t \in \triangle^{K-1}} s(w, t) = \frac{1}{2}\|Xw - Zt\|_2^2 \tag{3}$$

When considering the case $L > 1$, Eqn. (3) becomes $\min_{w \in \mathbb{R}^{D \times L}, t \in \triangle^{K-1}} \frac{1}{2}\|Xw - Zt\|_F^2$ where $Z \in \mathbb{R}^{N \times L \times K}$ and $\|\cdot\|_F$ is Frobenius norm. From Eqn. (2) and Eqn. (3), $s(w, t)$ is an approximation

---

**Algorithm 1** Alternating Minimization

1: **Input:** Model feature $X \in R^{N \times D}$; F-Label matrix $Z \in R^{N \times K}$; Learning stepsizes $\eta$ and $\beta$ for $w$ and $t$, respectively;
2: **Output:** Score of **WLSR**;
3: Initialize $t = \frac{1}{K}1_K$ and $w = \frac{1}{D}1_D$;
4: **while** $s$ not converge **do**
5:    $s = \frac{1}{2}\|Xw - Zt\|_2^2$ ;
6:    $w \leftarrow w - \eta X^T(Xw - Zt)$;
7:    **while** $t$ not converge **do**
8:      $t \leftarrow t - \beta Z^T(Zt - Xw)$;
9:      $t = \Pi_{\triangle^{K-1}}(t)$;    // Projection
10:    **end while**
11: **end while**
12: **Return:** $s$

**Algorithm 2** Fast Alternating Minimization

1: **Input:** Model feature $X \in R^{N \times D}$, F-Label matrix $Z \in R^{N \times K}$;
2: **Output:** Score of **WLSR**;
3: Initialize $t = \frac{1}{K}1_K$ and $w = \frac{1}{D}1_D$;
4: **while** $s$ not converge **do**
5:    $s = \frac{1}{2}\|Xw - Zt\|_2^2$;
6:    $w = (X^T X)^{-1}X^T Zt$;    // LSR for $w$
7:    $t = (Z^T Z)^{-1}Z^T Xw$;    // LSR for $t$
8:    $t = \text{Sparsemax}(t)$ ;    // Projection
9: **end while**
10: **Return:** $s$

---

of negative log-likelihood. Hence, a smaller $s(w, t)$ indicate the larger $T_m$ in Eqn. (1) and better transferability. We design an efficient algorithm to solve WLSR.

### 4.3 Fast Computation by Alternating Minimization

**Algorithm.** The optimization problem in Eqn. (3) can be formulated to a classical second-order conic program[42][44](simply called **SOCP**). However, the excessive data in our problem leads to a large dimension of the variable, making it inefficient for standard solvers. Therefore, we are motivated to find the smooth structure of the problem and design an alternating minimization algorithm to achieve fast computation. As shown in Algorithm 1 , we separately fix $w$ and $t$ to optimize the other one until the function value in Eqn. (3) converges. Specifically, when we fix $t$, the whole problem degenerates to a least square problem with respect to $w$. When we fix $w$, we also need to solve a least square problem concerning $t$ under the simplex constraint.

**Convergence Analysis.** We will prove the convergence property of the function value. Indeed, we prove a stronger condition that the function value decreases after each round of iterations on $w$ and $t$. From the monotone convergence theorem, the convergence can thus be derived. We first present the decreasing result of inner loop of $t$ by Theorem 1 and the same property holds for the update of $s$. Then the convergence of the whole algorithm can be derived by Theorem 2. The detailed proofs are placed in the Appendix Sec.A.

**Theorem 1.** *Suppose* $s(w,t) = \frac{1}{2}\|Xw - Zt\|_F^2$ *where* $X \in \mathbb{R}^{N \times D}$, $Z \in \mathbb{R}^{N \times K}$, $w \in \mathbb{R}^{D \times 1}$ *and* $t \in \triangle^{K-1}$, *the inner loop of* $t$ *in Algorithm 1 lines 7 - 10 decreases after each iteration. Specifically, denote* $\beta = 1/\|2Z^T Z\|$ *and* $t^+ = \Pi_{\triangle^{K-1}}(t - \beta\nabla s(w,t))$. *For any* $t \in \triangle^{K-1}$, $s(w,t^+) - s(w,t) \le -\frac{1}{2\beta}\|t - t^+\|^2 \le 0$.

**Theorem 2.** *Suppose* $s(w,t) = \frac{1}{2}\|Xw - Zt\|_2^2$ *where* $X \in \mathbb{R}^{N \times D}$, $Z \in \mathbb{R}^{N \times K}$, $w \in \mathbb{R}^{D \times 1}$ *and* $t \in \triangle^{K-1}$, *the function value in Algorithm 1 will converge. Specifically, denote* $w^\star, t^\star$ *as the result after one iteration of* $w, t$ *respectively, we have* $0 \le s(w^\star, t^\star) \le s(w^\star, t) \le s(w,t)$.

**Computational Speedup.** Although this algorithm 1 guarantees convergence, it is a bit time-consuming due to the two-level loop, we optimized this part and achieved similar results in very little time. Since the least squares solution is extremely fast, we performs least squares on $w$ and $t$, and then replace projection onto simplex with explicit Sparsemax transformation [45, 46], iteratively. The fast solver is illustrated in Algorithm 2 . we experimentally verify its convergence and find that the approach achieves impressive speedup.

## 5 Experiment

This section evaluates our method EMMS on different downstream tasks, including image classification, image caption, visual question answering, text question answering and referring expression comprehension. We put more experiments details in Appendix Sec.B. Moreover, we conduct a detailed ablation study to analyze our EMMS in Appendix Sec.C

### 5.1 Training Details

**Benchmark.** For **image classification**, We adopt 11 classification benchmarks , including FGVC Aircraft [47], Caltech-101 [48], Stanford Cars [49], CIFAR-10 [50], CIFAR-100 [50], DTD [51], Oxford 102 Flowers [52], Food-101 [53], Oxford-IIIT Pets [54], SUN397  [55], and VOC2007 [56]. For **image caption**, We use Flickr8k [57], Flickr30k [58], FlickrStyle10K-Humor [59], FlickrStyle10K-Romantic [59] and RSICD [60]. For **visual question answer**, We apply COCOQA [61], DAQUAR [62] and CLEVR [63]. For **text question answer** and **referring expression comprehension**, we separately use SQuAD1.1 [64] ,SQuAD2.0 [65] and RefCOCO [66], RefCOCO+ [66], RefCOCOg [67] .

**Ground truth.** In order to obtain the ground truth, we finetune all pre-trained models on all target datasets with a grid search of hyper-parameters. Details of target datasets and fine-tuning schemes are described in Appendix Sec.B.

**Evaluation protocol.** To assess how well a model selector predict the transferability of pre-trained models, we calculate the rank correlation between $\{T_m\}_{m=1}^M$ and $\{G_m\}_{m=1}^M$. Following the common practice [13, 31], we use *weighted Kendall's* $\tau_w$. The larger $\tau_w$ indicates a better correlation and better transferability metric. For computation complexity, we record the runtime of executing algorithm

Table 1: Comparison of different transferability metrics on ViT models regarding $\tau_w$ and the wall-clock time where EMMS(One) denotes EMMS with the one-hot label. Our proposed EMMS achieves the best transfer-ability assessment over 11 target tasks and exhibits higher efficiency than NLEEP.

| | Aircraft | Caltech | Cars | CF-10 | CF-100 | DTD | Flowers | Food | Pets | SUN | VOC | Avg. |
|---|---|---|---|---|---|---|---|---|---|---|---|---|
| | | | | | Weighted Kendall's tau $\tau_w$ | | | | | | | |
| LogME | 0.299 | 0.382 | 0.633 | **0.741** | 0.727 | 0.569 | 0.512 | 0.580 | 0.528 | 0.619 | 0.591 | 0.561 |
| NLEEP | -0.282 | 0.027 | 0.693 | 0.674 | 0.538 | 0.123 | -0.262 | 0.105 | 0.40 | 0.268 | 0.109 | 0.218 |
| TransRate | 0.244 | 0.412 | 0.487 | 0.260 | 0.702 | 0.533 | **0.655** | 0.542 | 0.707 | 0.612 | 0.651 | 0.527 |
| EMMS(One) | 0.412 | 0.444 | 0.565 | 0.740 | 0.736 | 0.621 | 0.562 | 0.579 | 0.740 | 0.592 | 0.730 | 0.611 |
| EMMS | **0.481** | **0.444** | **0.706** | 0.718 | **0.745** | **0.621** | 0.562 | **0.673** | **0.740** | **0.619** | **0.730** | **0.639** |
| | | | | | Wall-Clock Time (s) | | | | | | | |
| LogME | 8.93 | 10.89 | 30.28 | 53.07 | 62.13 | 4.78 | 9.27 | 104.92 | 6.28 | 425.43 | 7.42 | 65.76 |
| NLEEP | 553.7 | 716.8 | 1.1e3 | 8.0e3 | 1.2e4 | 183.7 | 819.2 | 3.4e4 | 256.4 | 2.7e4 | 288.3 | 7719.8 |
| TransRate | 19.43 | 19.21 | 36.9 | 61.73 | 63.82 | 8.73 | 18.26 | 110.79 | 15.51 | 89.92 | 5.11 | 40.85 |
| EMMS(One) | **4.12** | **4.45** | **8.07** | **19.45** | **26.18** | **2.65** | **4.03** | **39.72** | **3.50** | **24.84** | **4.07** | **12.82** |
| EMMS | 21.31 | 17.23 | 28.06 | 154.61 | 182.11 | 13.87 | 15.95 | 265.99 | 17.93 | 63.86 | 16.63 | 72.55 |

over all models given the feature and label on a target task and analyzed the computational complexity of EMMS as well as LogME. (Details can be found in Appendix Sec.C)

**Baseline.** For the image classification task, we choose NLEEP [31], TransRate [23], and LogME [13]as the baseline; for other multimodal tasks, we choose LogME with F-Label as the baseline; in addition, for the VQA task, we additionally compare PACTran [24]. Details of baselines and why we choose them are described in Appendix Sec.B.

## 5.2 Image Classification with ViT Models

Vision transformer [9] (ViT) models have been increasingly used for a variety of tasks and have achieved better results than CNN models. The architecture of ViT models are more complex than CNN models. Hence, how to do the model selection on ViT models is a more challenging and rewarding task. Details of pre-trained models are described in Appendix Sec.B.

**Performance and wall-clock time comparison.** As shown in Table.1 , our EMMS achieve the best average $\tau_w$ on 11 target datasets and the best $\tau_w$ on 9 target datasets with relatively short time. For example, EMMS outperforms LogME by 0.182 and 0.139 rank correlation $\tau_w$ on Aircraft, and VOC2007, respectively, showing the effectiveness of our EMMS in measuring the transfer-ability of pre-trained ViT models. On the other hand, for the remaining 2 target datasets (i.e. CF-10, DTD), our EMMS still has a marginal gap compared to the best-performing transferability metric. Besides, we find that the effect of model selection of EMMS in ViT models selection has an improvement compared to CNN models selection, we guess F-Label has spatial similarity with the model feature of ViT-base model because the foundation models are mostly transformer-based, which can model the relationship between model feature from Vit-base models and F-Labels more accurately.

## 5.3 Image Captioning

Here we treat image caption as a vocab-based classification task. That is we use a vocabulary and classify the caption into the index of some words in the vocabulary. Afterward, training is done according to the classification task criteria .Here we calculate the average $\tau_w$ and time of LogME with $K$ single F-label from $K$ foundation models we use respectively. We wants to select the best combination of image encoder and language encoder. Details of pre-trained models and the model architecture are described in Appendix Sec.B.

**Performance and wall-clock time comparison.** As shown in Table.2, EMMS is significantly ahead of baseline in both time and effect for each dataset. For example, EMMS outperforms LogME with the relative improvements of 39% and 37% in rank correlation $\tau_w$ on Flickr8k and Flickr30k, respectively. In addition, the time of EMMS is reduced by 83.7% and 79.8% relative to LogME on these two datasets, which shows the efficiency of our algorithm. The average rank correlation $\tau_w$ alone the five datasets is 0.64, which denotes EMMS has sufficient confidence.

Table 2: Comparison of different transferability metrics on image caption models in rank correlation $\tau_w$ with the ground truth and the wall-clock time. The LogME denotes using LogME with F-Label. Our proposed EMMS achieves the best transfer-ability assessment on each target task with much less time compared to LogME.

| | F8k | F30k | RSD | F10k-H | F10k-R | F8k | F30k | RSD | F10k-H | F10k-R |
|---|---|---|---|---|---|---|---|---|---|---|
| | Weighted Kendall's tau $\tau_w$ | | | | | Wall-Clock Time (s) | | | | |
| LogME | 0.483 | 0.368 | 0.501 | 0.780 | 0.654 | 425.67 | 1594.16 | 973.22 | 60.35 | 63.79 |
| EMMS | **0.660** | **0.504** | **0.704** | **0.802** | **0.678** | **69.01** | **321.32** | **88.77** | **16.56** | **14.59** |

## 5.4 Text Question Answering

For natural language understanding, we consider Text Question Answering (TQA) as a reading comprehension task, where the response to each question is a text segment extracted directly from the affiliated reading passage, or the question may indeed be deemed unanswerable. Details of pre-trained models and how to finetune are described in Appendix Sec.B.

**Performance and wall-clock time comparison.** In Table 3, the performance improvement of EMMS on the TQA is consistent with the enhancements observed in the earlier mentioned computer vision tasks. More specifically, our EMMS attains accuracies of 60.3% and 46.3% on the Stanford Question Answering Dataset (SQuAD) versions 1.1 and 2.0 respectively, using rank correlation $\tau_w$ as an evaluation metric. This represents a significant relative increment of 11.2% and 13.2% compared to the performance of LogME.

Table 3: Comparison of different transferability metrics on TQA models in rank correlation $\tau_w$ with the ground truth and the wall-clock time. The LogME denotes using LogME with F-Label.

Table 4: Comparison of different transferability metrics on referring expression models in rank correlation $\tau_w$ with ground truth and the time. The LogME denotes using LogME with F-Label.

| | SQu1.1 | SQu2.0 | SQu1.1 | SQu2.0 |
|---|---|---|---|---|
| | Weighted Kendall's tau $\tau_w$ | | Wall-Clock Time (s) | |
| LogME | 0.542 | 0.409 | 3587.22 | 3596.23 |
| EMMS | **0.603** | **0.463** | **571.23** | **589.78** |

| | Ref | Ref+ | Refg | Ref | Ref+ | Refg |
|---|---|---|---|---|---|---|
| | Weighted Kendall's tau $\tau_w$ | | | Wall-Clock Time (s) | | |
| LogME | 0.423 | 0.389 | 0.398 | 2457.87 | 2478.90 | 2298.76 |
| EMMS | **0.458** | **0.549** | **0.521** | **454.26** | **467.92** | **356.94** |

## 5.5 Referring Expression Comprehension

Referring expression comprehension (REC) is a widely challenging task because it requires precise alignment between linguistic concepts and image features. To address this, the objects in each image are represented as a sequence of discrete tokens, while their bounding box corner coordinates are turned into integer location tokens. This allows for a unified F-Label to be extracted using various language models. More details about the pre-trained models can be found in Appendix Sec.B.

**Performance and wall-clock time comparison.** As shown in Table 4, our EMMS continues to exhibit its superiority in the enhancement of performance on the REC task, an *instance-level cross-modal* localization task. Specifically, the proposed EMMS produces accuracies of 45.8%, 54.9%, and 52.1% on the RefCOCO, RefCOCO+, and RefCOCOg datasets respectively. This significantly surpasses its counterpart, LogME, in terms of margins when evaluated with rank correlation $\tau_w$.

## 5.6 Ablation Analysis

**Comparison with different number of F-Label** Here we denote the number of F-Label is $K$ and choose the image caption task to illustrate the impact of $K$ on our solution. As shown in Table 6. We find that increasing $K$ in a certain range brings a gain in effectiveness to our method, but when K becomes larger, the time also increases and we find that $K = 4$ is not as effective as $K = 3$. We believe that the increase in $K$ brings difficulties in fitting the true Label, resulting in a loss of effectiveness. Therefore, we use $K = 3$ for the sake of effect and time.

**Performance on F-Label using small model** On the one hand, using foundation model can extract the joint embedding compared to the small model, which allows EMMS to be extended to tasks with multiple forms of labels. On the other hand, the foundation model can handle many types of tasks,

Table 6: EMMS under different number of F-Label of transferability assessment on image caption task. The improvement of $K$ in a certain range brought an increase in rank correlation $\tau_w$.

| | F8k | F30k | RSD | F10k-H | F10k-R | | F8k | F30k | RSD | F10k-H | F10k-R |
|---|---|---|---|---|---|---|---|---|---|---|---|
| | Weighted Kendall's tau $\tau_w$ | | | | | | Weighted Kendall's tau $\tau_w$ | | | | |
| K=1 | 0.490 | 0.386 | 0.527 | 0.772 | 0.668 | K=2 | 0.574 | 0.454 | 0.553 | 0.762 | 0.646 |
| K=3 | **0.660** | **0.504** | **0.704** | **0.802** | **0.678** | K=4 | 0.660 | 0.504 | 0.704 | 0.802 | 0.644 |

so we can use the foundation model for different tasks for label embedding. As shown in Table 5, we experimentally demonstrate that the use of the foundation model leads to more accurate F-Label extraction and thus to an improvement in the performance of the method.

**The effect of using a single foundation model** We investigate how EMMS is influenced when only a single foundation model is provided. We conduct experiments on image classification and image captioning. We consider EMMS with the single foundation model including language foundation model (1) GPT-2 [27], (2) BERT [43] , (3) RoBerta [68], and multimodal foundation model (4) CLIP [26], (5) FLAVA [69], and (6) AltCLIP [70]. For comparison, we include the result of our EMMS with default setting (K=3, i.e. CLIP, BERT, and GPT-2) and the result of previous state-of-the-art methods obtained from LogME, NLEEP and TransRate. The results are reported in Table 20 and Table 5.

We have several observations. (1) Different downstream tasks prefer F-Labels obtained from different foundation models. No single foundation model is dominant in all target tasks. In particular, CLIP is not the best model for extracting F-Labels. (2) For image captioning, multimodal foundation models are more appropriate for extracting F-Labels than language foundation models. (3) Our EMMS can achieve the best results by combining F-Labels obtained from multiple foundation models.

Table 5: The effect of the single foundation model on EMMS. The results are obtained on image captioning regarding $\tau_w$.

| | F8k | F30k | RSD | F10kH | F10kR | Avg | SOTA/All |
|---|---|---|---|---|---|---|---|
| | Weighted Kendall's tau $\tau_w$ | | | | | | |
| LogME(Clip) | 0.530 | 0.393 | 0.618 | 0.764 | 0.634 | 0.588 | 0/5 |
| (1) Gpt2 | 0.566 | 0.393 | 0.431 | 0.715 | 0.618 | 0.545 | 0/5 |
| (2) Bert | 0.395 | 0.319 | 0.448 | **0.802** | **0.711** | 0.535 | 2/5 |
| (3) RoBerta | 0.346 | 0.111 | 0.587 | 0.571 | 0.566 | 0.436 | 0/5 |
| (4) $CLIP_B$ | 0.453 | 0.393 | **0.704** | **0.802** | 0.634 | 0.533 | 2/5 |
| (5) $CLIP_L$ | 0.510 | 0.448 | **0.704** | **0.802** | 0.678 | 0.628 | 2/5 |
| (6) FLAVA | 0.463 | 0.382 | 0.693 | 0.704 | 0.678 | 0.584 | 0/5 |
| (7) AltCLIP | 0.453 | 0.448 | 0.623 | **0.802** | 0.678 | 0.601 | 1/5 |
| EMMS | **0.660** | **0.504** | **0.704** | **0.802** | 0.678 | **0.670** | 4/5 |

# 6 Conclusion

How to select a pre-trained model for different tasks quickly and effectively is an important issue in the field of transfer learning. This paper proposes an efficient multi-task model selector(EMMS) that can be applied to many types of tasks. EMMS uses foundation model for Label embedding in order to transform diverse label formats of different tasks into the same form and see them as noisy labels. To estimate a model's transferability, EMMS model this problem as a simple weighted linear regression, which can be solved use an alternating minimization algorithm. Compared with existing methods, EMMS achieves the first model selection in multi-task scenarios, including image caption, referring segmentation, etc., with high speed and great results. For the **limitations** of the method, if the foundation model generalize very poor on downstream tasks, it may lead to low-quality label embedding, which is a drawback of our method. Moreover, building a holistic benchmark of various label embeddings would be useful in many applications such as multi-modal adaptation [71]. We leave it as a future work.

# 7 Acknowledgments

This paper is partially supported by the National Key RD Program of China No.2022ZD0161000, National Key RD Program of China(NO.2022ZD0160100) and the General Research Fund of Hong

Kong No.17200622. Besides, thanks Ruimao Zhang from CUHK(SZ) for thoughtful discussion and Prof. Anthony Man-Cho So from CUHK for his valuable discusion about solving the WLSR in the paper. At last, this work was done during the internship at Shanghai Artificial Intelligence Laboratory.

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

# A  Method

Here we derive in detail the regression with Unified Noisy Label Embeddings that appear in the method section of the text in Sec.A.1 and give complete proof of the convergence of the method in Sec.A.2.

## A.1  Regression with Unified Noisy Label Embeddings

**Setup.** we assume that label embedding $z$ is a linear mapping of the model feature with additive Gaussian noise with a variance of $\sigma_0^2$, as given by $z = z_0 + \epsilon = w^T \hat{x} + \epsilon$ and $\epsilon \sim N(0, \sigma_0^2 I_L)$ where $z_0 = w^T \hat{x}$ is the regression prediction, $w \in \mathbb{R}^{D \times L}$ and $\epsilon$ are regression weights and regression error, respectively, and $I_L$ is a L-by-L identity matrix.

We assume that F-labels $\{z_k\}_{k=1}^K$ obtained from different foundation models are oracles that independently provide noisy estimates of the label embedding $z$. Formally, we have $P(z_k|z) = N(z, \sigma_k^2 I_L)$. Without loss of generality, we assume that $L = 1$

Then the joint probability over noisy labels for a fixed $n$, That is, for given $x^n$, we have:

$$P(z_1^n, \cdots, z_K^n | x^n, w) = \int P(z_1^n, \cdots, z_K^n | z, x^n, w) P(z|x^n, w) dz \tag{4}$$

Due to the independence between $z_k$ and $x$, using the real label $z$, we can rewrite it as:

$$P(z_1^n, \cdots, z_K^n | x^n, w) = \int P(z_1^n, \cdots, z_K^n | z, w) P(z|x^n, w) dz \tag{5}$$

And using the independencies among $z_k$, we have:

$$P(z_1^n, \cdots, z_K^n | z, w) = \prod_{k=1}^K P(z_K^n | z, \sigma_1^2, \cdots, \sigma_k^2) = \frac{1}{(2\pi)^{\frac{K}{2}} \prod_{k=1}^K \sigma_k} \exp^{-\sum_{k=1}^K \frac{(z_k^n - z)^2}{2\sigma_k^2}} \tag{6}$$

Due to $P(z_k|z) = N(z, \sigma_k^2 I_L)$, we can rewrite it as :

$$P(z_1^n, \ldots, z_K^n | x^n, w) = \int \frac{1}{(2\pi)^{\frac{K+1}{2}} \prod_{k=0}^K \sigma_k} \exp^{-\sum_{k=1}^K \frac{(z_K^n - z)^2}{2\sigma_k^2} - \frac{(z - z_0)^2}{2\sigma_0^2}} dy \tag{7}$$

which can be calculated as :

$$P(z_1^n, \ldots, z_K^n | x^n, w) = A_1 \int e^{-A_2 y^2 + A_3^n y - A_4^n} dz = A_1 \sqrt{\frac{\pi}{A_2}} e^{\frac{(A_3^n)^2}{4A_2} - A_4^n} \tag{8}$$

where $A_1 = \prod_{k=0}^K 1/\sigma_k$, $A_2 = \sum_{k=0}^K 1/2\sigma_k^2$, $A_3^n = \sum_{k=0}^K z_k^n/\sigma_k^2$, and $A_4^n = \sum_{k=0}^K (z_k^n)^2/2\sigma_k^2$

Consider the joint probability over all $N$ instances, we have:

$$P(z_1^n, \ldots, z_K^n | X, w) = \prod_{i=1}^N A_1 \sqrt{\frac{\pi}{A_2}} e^{\frac{(A_3^n)^2}{4A_2} - A_4^n} \tag{9}$$

where $X \in R^{N \times D}$ denotes the feature matrix, $N$ is the number of data points and $D$ is the number of features.

Then given $N$ data points, the negative log-likelihood is given by

$$-\mathcal{L} = \underbrace{-N \log A_1 + \frac{N}{2} \log A_2}_{\mathcal{L}_1} + \underbrace{\frac{1}{2} \sum_{n=1}^N (A_4^n - \frac{(A_3^n)^2}{4A_2})}_{\mathcal{L}_2} + \text{const} \tag{10}$$

where $\mathcal{L}_1$ and $\mathcal{L}_2$ are given by

$$\mathcal{L}_1 = \frac{N}{2} \log \sum_{k=0}^K \frac{1}{2\sigma_k^2} + N \sum_{k=0}^K \log \sigma_k, \quad \mathcal{L}_2 = \sum_{n=1}^N \{ \sum_{k=0}^K \frac{(z_k^n)^2}{\sigma_k^2} - \frac{(\sum_{k=0}^K z_k^n/\sigma_k^2)^2}{\sum_{k=1}^K 1/\sigma_k^2} \} \tag{11}$$

Since $\mathcal{L}_1$ is independent of input data, we focus on $\mathcal{L}_2$. To simplify the notation, we re-denote $\gamma_k = 1/\sigma_k^2$ and $\Gamma = \sum_{k=1}^{K} \gamma_k$. Using this notation, $\mathcal{L}_2$ can be rearranged as:

$$\mathcal{L}_2 = \sum_{n=1}^{N} \{ \gamma_0 z_0^2 + \sum_{k=1}^{K} \gamma_k (z_k^n)^2 - \frac{(\sum_{k=1}^{K} \gamma_k z_k^n + \gamma_0 z_0)^2}{\Gamma + \gamma_0} \} \tag{12}$$

$$= \sum_{n=1}^{N} \{ (\gamma_0 - \frac{\gamma_0^2}{\Gamma + \gamma_0}) z_0^2 - (\frac{2\Gamma\gamma_0}{\Gamma + \gamma_0} \sum_{k=1}^{K} \frac{\gamma_0}{\Gamma} z_k^n) z_0 + \sum_{k=1}^{K} \gamma_k (z_k^n)^2 - (\sum_{k=1}^{K} \gamma_k z_k^n)^2 \} \tag{13}$$

$$= \sum_{n=1}^{N} \{ \frac{\Gamma\gamma_0}{\Gamma + \gamma_0} (z_0 - \sum_{k=1}^{K} \frac{\gamma_k}{\Gamma} z_k^n)^2 + \sum_{k=1}^{K} \gamma_k (z_k^n)^2 - (1 + \frac{\gamma_0}{\Gamma(\Gamma + \gamma_0)})(\sum_{k=1}^{K} \gamma_k z_k^n)^2 \} \tag{14}$$

$$= \sum_{n=1}^{N} \{ \frac{\Gamma\gamma_0}{\Gamma + \gamma_0} (w^T \hat{x}^n - \sum_{k=1}^{K} \frac{\gamma_k}{\Gamma} z_k^n)^2 + \sum_{k=1}^{K} \gamma_k (z_k^n)^2 - (1 + \frac{\gamma_0}{\Gamma(\Gamma + \gamma_0)})(\sum_{k=1}^{K} \gamma_k z_k^n)^2 \} \tag{15}$$

Hence, the negative likelihood in Eqn.(10 can be written as

$$-\mathcal{L} = \frac{\Gamma\gamma_0}{\Gamma + \gamma_0} \{ \underbrace{\frac{1}{2} \sum_{i=1}^{N} (w^T \hat{x}^n - \sum_{k=1}^{K} \frac{\gamma_k}{\Gamma} z_k^n)^2}_{s(w,t)} \} + \mathcal{R}(\gamma_k) \tag{16}$$

where $\mathcal{R}(\gamma_k) = \mathcal{L}_1 + \sum_{k=1}^{K} \gamma_k (z_k^n)^2 - (1 + \frac{\gamma_0}{\Gamma(\Gamma+\gamma_0)})(\sum_{k=1}^{K} \gamma_k z_k^n)^2$. The computational intractability of Eqn.(16) comes from the regularization term $\mathcal{R}(\gamma_k)$. Note that the coefficient $\frac{\Gamma\gamma_0}{\Gamma+\gamma_0} > 0$ and $\sum_{k=1}^{K} \frac{\gamma_k}{\Gamma} = 1$. By removing regularizer $\mathcal{R}(\gamma_k)$ and positive scale parameter $\frac{\Gamma\gamma_0}{\Gamma+\gamma_0}$, the minimization of negative log-likelihood can be approximately treated as a weighted linear square regression, as given by

$$\min_{w \in \mathbb{R}^{D \times 1}, t \in \triangle^{K-1}} s(w,t) = \frac{1}{2} \|Xw - Zt\|_2^2 \tag{17}$$

In Eqn.(17), $X \in \mathbb{R}^{N \times D}$ is the data matrix whose $n$-th row is model feature $(\hat{x}^n)^T$, $w \in \mathbb{R}^{D \times 1}$ are weight parameters, $Z \in \mathbb{R}^{N \times K}$ is F-Label matrix whose $k$-th column is the label embedding $z_k$, and $t \in \mathbb{R}^{K \times 1}$ satisfies that $1_K^T t = 1, t \geq 0$ which is a $(K-1)$-D simplex denoted as $\triangle^{K-1}$.

## A.2 Convergence Analysis and Proof Outline

We will prove the convergence property of the function value. Indeed, we demonstrate a stronger condition that the function value decreases after each round of iterations on $w$ and $t$. From the monotone convergence theorem, the convergence can thus be derived. For other convergence properties of alternating minimization, readers can refer to the literature [72], which can be of independent interest.

In the proof, we exploit the smoothness of the function and design a projection gradient descent method with sufficient decrease for the constraint optimization problem. The sufficient decrease in the unconstrained problem is a direct corollary.

**Definition 1.** *A function $f(x) : \mathbb{R}^d \to \mathbb{R}$ is said to be $\beta$-smooth with constant $\beta$ if*

$$|\nabla f(x) - \nabla f(y)| \leq \beta \|x - y\|, \forall x, y \in \mathbb{R}^d.$$

**Lemma 1.** *Suppose $X$ is the simplex constraint, and $y \in \mathbb{R}^d$, $\Pi$ denotes the projection operator. Then the inequality holds*

$$(\Pi_X(y) - x)^T (\Pi_X(y) - y) \leq 0.$$

*Proof.* For the projection $\Pi_X(y)$, it is a convex optimization problem and can be formulated as

$$\min_x f(x) = \|x - y\|_2^2,$$

where $x^T 1 = 1$ and $x > 0$. We denote $x^\star$ as the optimal solution to the problem. For the convex optimization problem, it holds for all $x \in \mathbb{R}^d$ that

$$\nabla f(x^\star)^T (x^\star - x) \leq 0.$$

Therefore we can derive

$$2(x^\star - y)^T (x^\star - x) \leq 0.$$

The lemma is proved. □

**Lemma 2.** *Let $f$ be a $\beta$-smooth function. For any $x, y \in \text{dom}(f)$*

$$\left| f(x) - f(y) - \nabla f(y)^T (x - y) \right| \leq \|x - y\|^2.$$

*Proof.*

$$\left| f(x) - f(y) - \nabla f(y)^T (x - y) \right| = \left| \int_0^1 \nabla f(y + t(x - y))^T (x - y) dt - \nabla f(y)^T (x - y) \right|$$

$$\leq \int_0^1 \|\nabla f(y + t(x - y)) - \nabla f(y)\| \|x - y\| dt$$

$$\leq \int_0^1 \beta t \|x - y\|^2 dt = \frac{\beta}{2} \|x - y\|^2.$$

The last inequality holds because $f$ is a $\beta$-smooth function. □

**Lemma 3.** *Suppose the function $f$ is the $\beta$-smooth function, and $X$ is the simplex constraint. For any $x, y \in X$, let $x^+ = \Pi_X(x - \frac{1}{\beta} \nabla f(x))$ and $g_X(x) = \beta(x - x^+)$. Then the inequality holds*

$$f(x^+) - f(y) \leq g_X(x)^T (x - y) - \frac{1}{2\beta} \|g_X(x)\|^2.$$

*Proof.* Using Lemma. 1, we have

$$(x^+ - (x - \frac{1}{\beta} \nabla f(x)))^T (x^+ - y) \leq 0.$$

which is equivalent to

$$\nabla f(x)^T (x^+ - y) \leq g_X(x)^T (x^+ - y).$$

By using Lemma. 2 and the fact $f(x^+) - f(y) = f(x^+) - f(x) + f(x) - f(y)$, we have

$$f(x^+) - f(y) \leq \nabla f(x)^T (x^+ - x) + \frac{\beta}{2} \|x^+ - x\|^2 + \nabla f(x)^T (x - y)$$

$$= \nabla f(x)^T (x^+ - y) + \frac{1}{2\beta} \|g_X(x)\|^2$$

$$\leq g_X(x)^T (x^+ - y) + \frac{1}{2\beta} \|g_X(x)\|^2$$

$$= g_X(x)^T (x^+ - x + x - y) + \frac{1}{2\beta} \|g_X(x)\|^2$$

$$= g_X(x)^T (x^+ - x) + g_X(x)^T (x - y) + \frac{1}{2\beta} \|g_X(x)\|^2$$

$$= g_X(x)^T (x - y) - \frac{1}{\beta} \|g_X(x)\|^2 + \frac{1}{2\beta} \|g_X(x)\|^2$$

$$= g_X(x)^T (x - y) - \frac{1}{2\beta} \|g_X(x)\|^2.$$

□

**Theorem 3.** *Suppose $s(w, t) = \frac{1}{2} \|Xw - Zt\|_F^2$ where $X \in \mathbb{R}^{N \times D}$, $Z \in \mathbb{R}^{N \times K}$, $w \in \mathbb{R}^{D \times 1}$ and $t \in \triangle^{K-1}$, the inner loop of $t$ in Algorithm lines 7 - 10 decreases after each iteration. Specifically, denote $\beta = 1/\|2Z^T Z\|$ and $t^+ = \Pi_{\triangle^{K-1}}(t - \beta \nabla s(w, t))$. For any $t \in \triangle^{K-1}$, $s(w, t^+) - s(w, t) \leq -\frac{1}{2\beta} \|t - t^+\|^2 \leq 0.$*

*Proof.* Since we fix parameter $w$ and consider the optimization problem of variable $t$, we denote $s(t) = s(w, t)$, which leads to $\nabla s(t) = -2Z^T(Xw^\star - Zt)$. For any $t_1, t_2 \in \text{dom}(s)$

$$\|\nabla s(t_1) - \nabla s(t_2)\| = \|2Z^T Z t_1 - 2Z^T Z t_2\| \leq \|2Z^T Z\|\|t_1 - t_2\|.$$

According to the definition 1, it shows that the $f(t)$ is $\beta$-smooth, where $\beta = \|2Z^T Z\|$. We denote $t \in \triangle^{K-1}$ to be the initial point and $t^+$ to be the result of one iteration of $t$, where $t^+ = \Pi_{\triangle^{K-1}}(t - \frac{1}{\beta}\nabla f(t))$. From Lemma 3, we can replace $x^+, y$ and $x$ with $t^+, t$, and $t$, repsectively. In this way, the inequality holds

$$0 \leq s(t^+) \leq s(t) - \frac{1}{2\beta}\|\beta(t - t^+)\|^2 \leq s(t)$$

$\square$

Therefore, according to **Monotone convergence theorem**, the function value of the iterative algorithm will converge.

**Theorem 4.** *Suppose $s(w, t) = \frac{1}{2}\|Xw - Zt\|_2^2$ where $X \in \mathbb{R}^{N \times D}$, $Z \in \mathbb{R}^{N \times K}$, $w \in \mathbb{R}^{D \times 1}$ and $t \in \triangle^{K-1}$, the function value in Algorithm will be convergent. Specifically, denote $w^\star, t^\star$ as the result after one iteration of $w, t$ respectively, we have $0 \leq s(w^\star, t^\star) \leq s(w^\star, t) \leq s(w, t)$.*

*Proof.* In the first step, we denote $t \in \triangle^{K-1}$ is the initial point, then use gradient descent algorithm to calculate $w^\star$. Since the optimization problem for $w$ is a convex optimization problem and use lemma 2, the decreasing property for the gradient part can be derived. That is, for each $w \in \mathbb{R}^{D \times 1}$, we have $s(w^\star, t) \leq s(w, t)$. In the second step, we fix $w$ as $w^\star$, from Theorem 3, we have $s(w^\star, t^\star) \leq s(w^\star, t)$. Therefore, the value of $s(w, t)$ satisfies: $0 \leq s(w^\star, t^\star) \leq s(w^\star, t) \leq s(w, t)$, from **Monotone convergence theorem**, $s(w, t)$ converges to the limiting point. As shown above, the overall convergence of our algorithm is guaranteed. $\square$

# B  Experiment

In this section, we present detailed descriptions of datasets in Sec. B.2, pre-trained models and baselines in Sec. B.3, and ground-truth scores in Sec. B.4 in various target tasks. More ablation studies can be found in Sec. C.

**Foundation Models.** On image classification, image captioning, referring expression comprehension, and visual question answering, we use foundation models CLIP [26], BERT [43] and GPT-2 [27]. On text question answering, we use foundation models GPT-2 [27], BART [73], and ELECTRA [74]. CLIP was trained on a large dataset of images and their corresponding captions, which can understand the relationship between images and text. BERT is a pre-trained language model that can understand and generate natural language. GPT-2 was trained on a large corpus of text and can be fine-tuned for specific tasks such as text completion and text summarization. Bart is a sequence-to-sequence model, which is both auto-regressive and bidirectional. Electra is a different type of language model that key idea is to pre-train a generator model to produce fake data and shows promising results in various NLP tasks.

**Interpretation of weighted Kendall's tau.** The Kendall's $\tau$ represents the ratio of concordant pairs minus discordant pairs when enumerating all pairs of $\{T_m\}_{m=1}^M$ and $\{G_m\}_{m=1}^M$ as given by

$$\tau = \frac{2}{M(M-1)} \sum_{1 \leq i < j \leq M} \text{sgn}(G_i - G_j)\text{sgn}(T_i - T_j) \tag{18}$$

where $\text{sgn}(x)$ returns $-1$ if $x < 0$ and $1$ otherwise. In this work, a weighted version of Kendall's $\tau$, denoted as $\tau_w$, is employed to assess transferability metrics considering that a top-performing model is always preferred for target tasks in transfer learning. In principle, a larger $\tau_w$ implies the transferability metric can rank pre-trained models better. And if a metric can rank top-performing models better, $\tau_w$ would be also larger. We also use other measurements to assess the performance of transferability metrics in Table 17 of Sec. C.

Table 7: Comparison of different transferability metrics on CNN models regarding $\tau_w$ and the wall-clock time where EMMS(One) denotes EMMS with the one-hot label. Our proposed EMMS achieves the best transfer-ability assessment over 11 target tasks and exhibits higher efficiency than NLEEP.

| | Aircraft | Caltech | Cars | CF-10 | CF-100 | DTD | Flowers | Food | Pets | SUN | VOC | Avg. |
|---|---|---|---|---|---|---|---|---|---|---|---|---|
| | Weighted Kendall's tau $\tau_w$ | | | | | | | | | | | |
| LEEP | -0.234 | 0.605 | 0.367 | 0.824 | 0.677 | 0.486 | -0.243 | 0.491 | 0.389 | 0.722 | 0.371 | 0.409 |
| LogME | 0.506 | 0.435 | **0.576** | 0.852 | 0.677 | 0.647 | 0.111 | 0.385 | 0.411 | 0.487 | 0.669 | 0.509 |
| NLEEP | -0.41 | **0.614** | 0.265 | 0.818 | 0.805 | **0.796** | 0.122 | 0.214 | **0.753** | **0.925** | 0.687 | 0.611 |
| TransRate | 0.172 | 0.269 | 0.172 | 0.513 | 0.197 | 0.336 | -0.176 | -0.071 | 0.173 | 0.612 | 0.651 | 0.236 |
| EMMS(One) | 0.481 | 0.546 | 0.304 | 0.963 | 0.804 | 0.701 | 0.498 | 0.588 | 0.574 | 0.638 | 0.707 | 0.618 |
| EMMS | **0.556** | 0.562 | 0.565 | **0.963** | **0.840** | 0.720 | **0.498** | **0.608** | 0.604 | 0.667 | **0.735** | **0.664** |
| | Wall-Clock Time (s) | | | | | | | | | | | |
| LEEP | 5.1 | 4.9 | 8.3 | 22.3 | 23.8 | 3.5 | 3.8 | 37.1 | 3.9 | 21.1 | 4.8 | 10.4 |
| LogME | 30.36 | 31.24 | 56.26 | 90.34 | 188.3 | 15.16 | 22.27 | 334.53 | 17.55 | 180.01 | 20.05 | 289.64 |
| NLEEP | 253.8 | 488.7 | 973.8 | 1.1e4 | 1.7e4 | 146.0 | 294.0 | 2.0e4 | 580.8 | 8.6e3 | 678.8 | 5455.9 |
| TransRate | 147.90 | 163.41 | 300.29 | 65.25 | 193.64 | 75.48 | 166.24 | 195.92 | 60.53 | 430.33 | 18.72 | 165.24 |
| EMMS(One) | 17.43 | 20.53 | 35.22 | 70.01 | 78.24 | 12.75 | 18.04 | 116.23 | 15.04 | 70.98 | 18.42 | 42.99 |
| EMMS | 65.85 | 63.49 | 79.79 | 245.49 | 295.37 | 46.38 | 63.52 | 417.80 | 59.64 | 173.59 | 64.60 | 143.2 |

Table 8: Comparison of different transferability metrics on VQA models in rank correlation $\tau_w$ with the ground truth and the wall-clock time. The LogME denotes using LogME with F-Label. Our proposed EMMS performs better than PACTran head over 3 target tasks with much less time.

| | DAQUAR | COCO-QA | CLEVR | DAQUAR | COCO-QA | CLEVR |
|---|---|---|---|---|---|---|
| | Weighted Kendall's tau $\tau_w$ | | | Wall-Clock Time (s) | | |
| LogME | 0.586 | 0.591 | 0.281 | 116.72 | 716.35 | 4665.06 |
| PACTran(Dir) | 0.671 | 0.296 | 0.347 | 633.16 | 1169.91 | 428.03 |
| PACTran(Gam) | 0.595 | 0.419 | 0.319 | 614.23 | 1061.72 | 428.49 |
| PACTran(Gau) | 0.478 | 0.378 | 0.415 | 637.39 | 1075.88 | 418.34 |
| EMMS | **0.712** | **0.812** | **0.804** | **50.54** | **263.72** | **274.56** |

## B.1 More experimental results

### B.1.1 Performance on Image Classification with CNN Models

**Performance and wall-clock time comparison.** We compare EMMS with previous LEEP, NLEEP, LogME, and TransRate. As shown in Table.7, our EMMS achieve the best average $\tau_w$ on 11 target datasets and the best $\tau_w$ on 6 target datasets. Compared to NLEEP, which is the most effective other than EMMS, we have almost 1/40 of the time of NLEEP.

### B.1.2 Performance on Visual Question Answering

To further demonstrate the generality of EMMS in multi-model tasks, we show how EMMS can work for VQA. We follow previous practice ( [24]) which treats VQA as a classification task (vocab-based VQA). That is, we construct a vocabulary based on the top answers in the training sets and classify them into some of those labels. The models to be selected and the architecture is the same as in the image captioning .

**Performance and wall-clock time comparison.** As shown in Table.8, EMMS is clearly ahead of PACTran in terms of results and time, proving that EMMS has the ability to handle multi-modal tasks very well. We can find that EMMS outperforms PACTran on all datasets. In particular, EMMS achieves 93.8% and 93.7% gain over PACTran on the COCO-QA and CLEVR datasets with rank correlation $\tau_w$ while reducing time consumption by 75.1% and 34.3% respectively compared to Pactran. This indicates that EMMS performs well on both ordinary VQA datasets(DAQUAR, COCO-QA) as well as VQA datasets(CLEVR) that focus on inference capabilities.

### B.1.3 Regression

In addition to image classification and a variety of multi-modal tasks, here we show that EMMS can also be used for regression tasks. The daatasets for regression task we use is CUB200 [75] and IIIT

Pets [54]. The input is an image containing various birds and pets, respectively. We need to predict the coordinates of the bird's or pet's bounding box in the image and mean square error (MSE) on the test data is the ground-truth. The pre-trained models used are the same as the image classification task with CNN models and the only baseline is LogME. We extract F-Labels using Bert and RoBerta.

As shown in Table 9, EMMS significantly outperforms LogME, with 29.5% and 13.9% performance improvement on CUB and Pets respectively.

Table 9: Comparison of different transferability metrics on regression models in rank correlation $\tau_w$ with ground truth.

|  | CUB | Pets |
|---|---|---|
|  | Weighted Kendall's tau $\tau_w$ | |
| LogME | 0.464 | 0.437 |
| EMMS | **0.601** | **0.498** |

## B.2 Descriptions of Datasets

### B.2.1 Image Classification

For image classification, we adopt 11 classification benchmarks , including FGVC Aircraft [47], Caltech-101 [48], Stanford Cars [49], CIFAR-10 [50], CIFAR-100 [50], DTD [51], Oxford 102 Flowers [52], Food-101 [53], Oxford-IIIT Pets [54], SUN397 [55], and VOC2007 [56]. These datasets cover a broad range of classification tasks, which include scene, texture, and coarse/fine-grained image classification, which are widely used in transfer learning. In particular, CF10 and VOC2007 are typical coarse-grained classification datasets, Aircraft, and Cars are typical fine-grained classification datasets, and CF100 contains both coarse- and fine-grained classifications.

### B.2.2 Image Captioning

For image captioning, We use Flickr8k [57], Flickr30k [58], FlickrStyle10K-Humor [59], FlickrStyle10K-Romantic [59] and RSICD [60]. Among them, Flickr8k and Flickr30k have commonly used image captioning datasets for natural images and have no emotional color; RSICD is a commonly used image captioning dataset in remote sensing; Flickr10k-H and Flickr10k-R are also image captioning datasets for natural images, but their images are depicted with humorous and romantic emotional colors, respectively.

### B.2.3 Visual Question Answering

For visual question answering, we apply COCOQA [61], DAQUAR [62] and CLEVR [63].Among them, DAQUAR is an early VQA dataset on real images; CLEVR is a synthetic dataset, which is a visual scene composed of some simple geometric shapes, focusing on evaluating the inference ability of VQA models; the questions and answers of COCO-QA are generated by NLP algorithms, and the images are from the COCO dataset, which is also a commonly used VQA dataset.

### B.2.4 Text Question Answering

For text question answering, we separately use SQuAD1.1 [64] ,SQuAD2.0 [65], which are collections of question-answer pairs derived from Wikipedia articles and are widely used in text question answer.

### B.2.5 Referring Expression Comprehension

For referring expression comprehension, we separately use RefCOCO [66], RefCOCO+ [66] and RefCOCOg [67].Specifically, RefCOCO includes instances where there is only one object of its kind in the image, while RefCOCO+ includes instances where multiple objects of the same kind exist in the image.

### B.3 Pre-trained Models and Baselines

#### B.3.1 Image Classification

**Pre-trained Models.** For **CNN-based** models, We select 11 widely-used CNN models including ResNet-34 [6], ResNet-50 [6], ResNet-101 [6], ResNet-152 [6], DenseNet-121 [7], DenseNet-169 [7], DenseNet-201 [7], MNet-A1 [76], MobileNetV2 [77], GoogleNet [8], and InceptionV3 [78]. All these models are trained on ImageNet dataset [1], which are widely used within the field of migration learning. For **ViT-based** models, we collect 10 ViT models including ViT-T [9], ViT-S [9], ViT-B [9], DINO-S [79], MoCov3-S [80] , PVTv2-B2 [11], PVT-T [11], PVT-S [11], PVT-M [11], and Swin-T [10], which are widely used in various vision tasks. Besides, we append EMMS with one-hot label, which degenerates to a linear regression whose label is the one-hot vector. We fine-tune these models on the 11 target datasets to obtain the ground truth.

**Comparison Baselines.** Here we use some of the latest methods as baselines, including LEEP [12], NLEEP [31], LogME [13], and TransRate [23], which have been experimented with model selection on image classification tasks.

#### B.3.2 Image Captioning

**Pre-trained Models.** We use a classic and effective image captioning model architecture, which contains an image encoder and a language encoder to extract the features of the image and the corresponding caption, then fuses the image feature and the text feature and input it to the classifier. We aim to choose the best combination of image encoder and language encoder. Besides, We finetune each model in COCO Caption [81] and use these as the pre-trained models.

Specifically, We separately use ViT-B [9],Swin-B [10], Swinv2-B [82] as image encoder and Bert [43], Roberta [68], Bart [73] as language encoder, and use VisionEncoderDecoderModel from HuggingFace as the model architecture. Following the setting in PACTran [24], We finetune the model in COCO Caption [81] and use these as the pre-trained models. Following common practice( [83]) , we treat image captioning as a vocab-based classification task. That is we use a vocabulary and classify the caption into the index of some words in the vocabulary. Afterward, training is done according to the classification task criteria.

**Comparison Baselines.** In this common setup, each caption is converted to a matrix $Y \in R^{L \times N}$, where $L$ denotes the length of the caption after padding or truncation and $N$ denotes the size of the vocabulary, and each row in the matrix is a one-hot vector. Since $N$ is generally very large, Existing model selection metrics do not scale to this case due to the huge amount of time spent. The only baseline we use is to model the fused feature with F-label using LogME since only LogME can handle the regression task. Here we calculate the average $\tau_w$ and time of it with $K$ single F-label from $K$ foundation models we use respectively.

#### B.3.3 Visual Question Answering

**Pre-trained Models.** The model architecture and the model selection settings are the same as in the image captioning, Following the setting in PACTran [24], here we use the model after finetune on VQA-v2 [83] as the pre-trained model waiting for selection and treat VQA as a vocab-based classification task.

**Comparison Baselines.** Here we calculate the average $\tau_w$ and time of it with $K$ single F-label from $K$ foundation models we use respectively. And in addition to that, the three methods proposed in PACTran [24] are added here, which are the only methods currently applied to VQA tasks.

#### B.3.4 Text Question Answering

**Pre-trained Models.** The selected models include BERT-Large [43], RoBERTa-Large [68], XLNet-Large [84], DeBERTa [85] (XLarge), DeBERTa-V2 [85] (XLarge and XXLarge), DeBERTa-V3 [86] (Base, Small, XSmall). More specifically, we simultaneously input the question and passage into the aforementioned models, utilizing the distinctive symbol [SEP] to demarcate them. By stacking the predicted head onto each model, we could further fine-tune the model such that it can predict the start and end positions of the answer within the passage. This is achieved by using two binary classifiers, where one is dedicated to identifying the start position and the other to pinpointing the end.

Table 10: The fine-tuning accuracy of supervised CNN models on 11 target tasks.

| | Aircraft | Caltech | Cars | CF-10 | CF-100 | DTD | Flowers | Food | Pets | SUN | VOC |
|---|---|---|---|---|---|---|---|---|---|---|---|
| ResNet-34 | 84.06 | 91.15 | 88.63 | 96.12 | 81.94 | 72.96 | 95.2 | 81.99 | 93.5 | 61.02 | 84.6 |
| ResNet-50 | 84.64 | 91.98 | 89.09 | 96.28 | 82.8 | 74.72 | 96.26 | 84.45 | 93.88 | 63.54 | 85.8 |
| ResNet-101 | 85.53 | 92.38 | 89.47 | 97.39 | 84.88 | 74.8 | 96.53 | 85.58 | 93.92 | 63.76 | 85.68 |
| ResNet-152 | 86.29 | 93.1 | 89.88 | 97.53 | 85.66 | 76.44 | 96.86 | 86.28 | 94.42 | 64.82 | 86.32 |
| DenseNet-121 | 84.66 | 91.5 | 89.34 | 96.45 | 82.75 | 74.18 | 97.02 | 84.99 | 93.07 | 63.26 | 85.28 |
| DenseNet-169 | 84.19 | 92.51 | 89.02 | 96.77 | 84.26 | 74.72 | 97.32 | 85.84 | 93.62 | 64.1 | 85.77 |
| DenseNet-201 | 85.38 | 93.14 | 89.44 | 97.02 | 84.88 | 76.04 | 97.1 | 86.71 | 94.03 | 64.57 | 85.67 |
| MNet-A1 | 66.48 | 89.34 | 72.58 | 92.59 | 72.04 | 70.12 | 95.39 | 71.35 | 91.08 | 56.56 | 81.06 |
| MobileNetV2 | 79.68 | 88.64 | 86.44 | 94.74 | 78.11 | 71.72 | 96.2 | 81.12 | 91.28 | 60.29 | 82.8 |
| Googlenet | 80.32 | 90.85 | 87.76 | 95.54 | 79.84 | 72.53 | 95.76 | 79.3 | 91.38 | 59.89 | 82.58 |
| InceptionV3 | 80.15 | 92.75 | 87.74 | 96.18 | 81.49 | 72.85 | 95.73 | 81.76 | 92.14 | 59.98 | 83.84 |

Table 11: The fine-tuning accuracy of vision transformer models on 11 target tasks.

| | Aircraft | Caltech | Cars | CF-10 | CF-100 | DTD | Flowers | Food | Pets | SUN | VOC |
|---|---|---|---|---|---|---|---|---|---|---|---|
| ViT-T | 71.26 | 89.39 | 82.09 | 96.52 | 81.58 | 71.86 | 95.5 | 81.96 | 91.44 | 58.4 | 83.1 |
| ViT-S | 73.12 | 92.7 | 86.72 | 97.69 | 86.62 | 75.08 | 96.79 | 86.26 | 94.02 | 64.76 | 86.62 |
| ViT-B | 78.39 | 93.47 | 89.26 | 98.56 | 89.96 | 77.66 | 97.98 | 88.96 | 94.61 | 68.62 | 87.88 |
| PVTv2-B2 | 84.14 | 93.13 | 90.6 | 97.96 | 88.24 | 77.16 | 97.89 | 88.67 | 93.86 | 66.44 | 86.44 |
| PVT-T | 69.76 | 90.04 | 84.1 | 94.87 | 75.26 | 72.92 | 95.8 | 83.78 | 91.48 | 61.86 | 84.6 |
| PVT-S | 75.2 | 93.02 | 87.61 | 97.34 | 86.2 | 75.77 | 97.32 | 86.98 | 94.13 | 65.78 | 86.62 |
| PVT-M | 76.7 | 93.75 | 87.66 | 97.93 | 87.36 | 77.1 | 97.36 | 85.56 | 94.48 | 67.22 | 87.36 |
| Swin-T | 81.9 | 91.9 | 88.93 | 97.34 | 85.97 | 77.04 | 97.4 | 86.67 | 94.5 | 65.51 | 87.54 |
| MoCov3-S | 76.04 | 89.84 | 82.18 | 97.92 | 85.84 | 71.88 | 93.89 | 82.84 | 90.44 | 60.6 | 81.84 |
| DINO-S | 72.18 | 86.76 | 79.81 | 97.96 | 85.66 | 75.96 | 95.96 | 85.69 | 92.59 | 64.14 | 84.8 |

**Comparison Baselines.** Here we calculate the average $\tau_w$ and time of it with F-labels from $K$ foundation models respectively.

### B.3.5 Referring Expression Comprehension

**Pre-trained Models.** The candidate multi-modal architectures considered for REC task incorporate Blip [87], ALBEF [88], CLIP [26] (ViT-B-32, ViT-B-16, ViT-L-14, ViT-L-14-336, RN50), OFA [89] (Base, Large, Huge). In practice, we respectively extract the visual and textual representations from each of these models and feed them into a multi-modal interaction module followed by a stacked detection head, and further fine-tune the model to generate the ground truth of model selection.

**Comparison Baselines.** Here we calculate the average $\tau_w$ and time of LogME with $K$ single F-label from $K$ foundation models we use respectively.

### B.4 Fine-tuning Score on Various Target Tasks

### B.4.1 Image Classification

**Fine-tuning Details.** The ground truth of the problem of pre-trained model ranking is to fine-tune all pre-trained models with a hyper-parameters sweep on target datasets. Given the model and the target dataset, two of the most important parameters would be learning rate and weight decay in optimizing the model [90]. Therefore, we carefully fine-tune pre-trained models with a grid search of learning rate in $\{1e-1, 1e-2, 1e-3, 1e-4\}$ and weight decay in $\{1e-3, 1e-4, 1e-5, 1e-6, 0\}$. And using SGD optimizer. After determining the best hyper-parameters candidate, we fine-tune the pre-trained model on the target dataset with the candidate and then obtain the test accuracy as the ground truth. We use a Tesla V100 with a batch size of 128 to perform finetuning. All input images are resized to $224 \times 224$. To avoid random error, we repeat the above fine-tuning procedure three times and take an average to obtain the final fine-tuning accuracy. For reference, we list the fine-tuning accuracy of supervised CNN models in Table.10, and vision transformer models in Table 11, respectively.

### B.4.2 Image Captioning and Visual Question Answering

**Fine-tuning Details.** The setting of finetune here is approximately the same as in image classification. We carefully fine-tune pre-trained models with a grid search of learning rate in $\{1e-4, 1e-5, 1e-6\}$ and weight decay in $\{1e-4, 1e-5, 1e-6\}$. And using AdamW optimizer. After determining the best hyper-parameters candidate, we fine-tune the pre-trained model on the target dataset with the candidate and then obtain the test BLEU-4 and accuracy as the ground truth. However, since Flickr10k-H and Flickr10k-R do not provide a test set, we use a 6:1 ratio to divide the original training set of 7000 images into a training set and a test set. For visual question answering, Due to the lack of a test set for CLEVR dataset, we also assign its training set as training set and test set in the ratio of 6:1. We use an Nvidia A100 with a batch size of 64 to perform finetuning. All input images are resized to $224 \times 224$. To avoid random error, we repeat the above fine-tuning procedure three times and take an average to obtain the final fine-tuning accuracy. For inference, We use BLEU-4 as the score for the model with image captioning and accurarcy as the score for the model with VQA. we list result of image captioning models in Table.12, and visual question answering models in Table 13, respectively.

Table 12: The fine-tuning BLEU-4 of image captioning models on 5 target tasks.

|  | F8k | F30k | RSD | F10k-H | F10k-R |
|---|---|---|---|---|---|
| Vit-Bert | 18.51 | 26.65 | 31.39 | 5.31 | 5.18 |
| Vit-Roberta | 20.53 | 23.70 | 29.92 | 5.88 | 5.48 |
| Vit-Bart | 21.90 | 25.13 | 31.35 | 5.75 | 5.53 |
| Swinvit-Bert | 22.91 | 26.61 | 33.54 | 6.24 | 5.67 |
| Swinvit-Roberta | 23.99 | 28.84 | 33.07 | 7.11 | 5.49 |
| Swinvit-Bart | 24.68 | 28.03 | 32.99 | 6.10 | 5.95 |
| Swin2vit-Bert | 25.69 | 31.33 | 35.45 | 5.86 | 5.49 |
| Swin2vit-Roberta | 23.40 | 28.81 | 36.22 | 6.80 | 7.13 |
| Swin2vit-Bart | 26.24 | 30.35 | 34.72 | 7.90 | 5.96 |

Table 13: The fine-tuning accuracy of visual question answering models on 3 target tasks.

|  | DAQUAR | COCO-QA | CLEVR |
|---|---|---|---|
| Vit-Bert | 25.01 | 55.11 | 59.29 |
| Vit-Roberta | 26.38 | 57.30 | 62.80 |
| Vit-Bart | 26.30 | 59.60 | 64.98 |
| Swinvit-Bert | 28.05 | 61.72 | 68.25 |
| Swinvit-Roberta | 27.75 | 62.81 | 66.09 |
| Swinvit-Bart | 27.06 | 60.62 | 67.17 |
| Swin2vit-Bert | 26.45 | 63.1 | 67.4 |
| Swin2vit-Roberta | 26.33 | 66.54 | 65.91 |
| Swin2vit-Bart | 26.25 | 64.4 | 70.34 |

### B.4.3 Text Question Answering

**Fine-tuning Details.** The accuracy of most models in TQA is provided by DeBERTa [85, 86], except for DeBERTa-V3 [86](Base, Small, XSmall). Following the setting of Bert [43], we finetune these models with a batch size of 24 for 2 epochs. We use AdamW optimizer with an initial learning rate of $3e-5$, polynomial decay. The Dev F1 score is used for pre-trained model ranking. All experiments are implemented on an NVIDIA Tesla A100 GPU. The finetune accuracy is shown in Table 14.

Table 14: The standard metric the Dev F1 score of text question answering models on 2 target tasks.

|  | SQu1.1 | SQu2.0 |
|---|---|---|
| BERT-Large | 90.9 | 81.8 |
| RoBERTa-Large | 94.6 | 89.4 |
| XLNet-Large | 95.1 | 90.6 |
| DeBERTa-Large | 95.5 | 90.7 |
| DeBERTa-V2-XLarge | 95.8 | 91.4 |
| DeBERTa-V2-XXLarge | 96.1 | 92.2 |
| DeBERTa-V3-Base | 93.9 | 88.4 |
| DeBERTa-V3-Small | 89.8 | 82.9 |
| DeBERTa-V3-XSmall | 91.5 | 84.8 |

Table 15: The standard metric Acc@0.5 of referring expression comprehension models on 3 target tasks.

|  | RefCOCO | RefCOCO+ | RefCOCOg |
|---|---|---|---|
| Blip | 88.67 | 84.68 | 85.08 |
| ALBEF | 87.98 | 82.20 | 82.89 |
| CLIP-ViT-B-32 | 83.20 | 74.56 | 76.98 |
| CLIP-ViT-B-16 | 87.35 | 80.12 | 81.69 |
| CLIP-ViT-L-14 | 90.17 | 86.09 | 87.13 |
| CLIP-ViT-L-14-336 | 91.67 | 87.60 | 87.89 |
| CLIP-RN50 | 84.69 | 76.72 | 79.39 |
| OFA-Base | 88.48 | 81.39 | 82.29 |
| OFA-Large | 90.05 | 85.80 | 85.89 |
| OFA-Huge | 92.04 | 87.86 | 88.07 |

### B.4.4 Referring Expression Comprehension

**Fine-tuning Details.** For referring expression comprehension, the standard metric Acc@0.5 on the validation set is used as the ground truth. For finetuning, we use a batch size of 128 with a resolution of $512 \times 512$ for each image. We finetune the models on each dataset for 12 epochs with a learning rate of $\{3e-5, 5e-5\}$ and weight decay in $\{1e-3, 1e-5\}$ using Adam optimizer. The best performance on the validation set for each task is reported among these hyper-parameters. Table 15 shows the performance of referring expression comprehension models.

Table 16: The fine-tuning MSE on test set of models used in regression on 2 target tasks.

|  | CUB | Pets |
| --- | --- | --- |
| ResNet-34 | $4.114e-4$ | $4.245e-5$ |
| ResNet-50 | $3.521e-4$ | $4.489e-5$ |
| ResNet-101 | $2.746e-4$ | $3.224e-5$ |
| ResNet-152 | $2.539e-4$ | $2.775e-5$ |
| DenseNet-121 | $5.354e-4$ | $1.096e-4$ |
| DenseNet-169 | $4.787e-4$ | $9.469e-5$ |
| DenseNet-201 | $4.651e-4$ | $1.058e-4$ |
| MNet-A1 | $1.1475e-3$ | $1.878e-4$ |
| MobileNetV2 | $6.253e-4$ | $9.510e-5$ |
| Googlenet | $7.192e-4$ | $1.197e-4$ |
| InceptionV3 | $6.174e-4$ | $9.633e-5$ |

Table 17: EMMS under different measurements of transferability assessment. The results are obtained on Flickr8k and RSICD datasets with image captioning task and Aircraft and DTD datasets with image classification task with ViT-based models. EMMS outperforms LogME and other baselines under various measures.

| Data | Method | Rel@1 | Rel@3 | $r$ | $r_w$ | $\tau$ | $\tau_w$ | Data | Method | Rel@1 | Rel@3 | $r$ | $r_w$ | $\tau$ | $\tau_w$ |
| --- | --- | --- | --- | --- | --- | --- | --- | --- | --- | --- | --- | --- | --- | --- | --- |
| 2*F8k | LogME | 0.928 | **1.0** | 0.735 | 0.799 | 0.537 | 0.483 | 2*RSD | LogME | 0.957 | **1.0** | 0.727 | 0.708 | 0.518 | 0.501 |
|  | EMMS | **1.0** | **1.0** | **0.741** | **0.823** | **0.667** | **0.660** |  | EMMS | **1.0** | **1.0** | **0.783** | **0.765** | **0.611** | **0.705** |
| 3*Aircraft | LogME | 0.852 | 0.993 | 0.407 | 0.060 | 0.378 | 0.299 | 3*DTD | LogME | 0.992 | **1.0** | 0.641 | 0.694 | 0.556 | 0.569 |
|  | TransRate | **0.926** | **0.967** | 0.457 | 0.499 | 0.289 | 0.244 |  | TransRate | **0.992** | **1.0** | 0.607 | 0.676 | 0.422 | 0.533 |
|  | EMMS | **0.926** | **0.967** | **0.622** | **0.608** | **0.511** | **0.481** |  | EMMS | **0.992** | **1.0** | **0.704** | **0.785** | **0.644** | **0.621** |

### B.4.5 Regression

**Fine-tuning Details.** For regression, mean square error (MSE) on the test data is the ground truth. For finetuning, we use a batch size of $64$ with resolution of $224 \times 224$ for each image. we carefully fine-tune pre-trained models with a grid search of learning rate in $\{1e-1, 1e-2, 1e-3, 1e-4\}$ and weight decay in $\{1e-3, 1e-4, 1e-5, 1e-6, 0\}$ with SGD optimizer. The fine-tuning MSE on test set of models used in regression is in Table 16

Table 18: The effect of Label Embedding in EMMS. Three variants of EMMS are considered: (1) EMMS with one-hot label; (2) EMMS with single F-Label; (3) EMMS with multiple F-Labels which is the original. We see that label embedding brings some performance improvement to EMMS.

|  | Aircraft | Caltech | Cars | CF-10 | CF-100 | DTD | Flowers | Food | Pets | SUN | VOC | Avg. |
| --- | --- | --- | --- | --- | --- | --- | --- | --- | --- | --- | --- | --- |
|  | | | | | Weighted Kendall's tau $\tau_w$ | | | | | | | |
| (1) | 0.481 | 0.546 | 0.304 | 0.963 | 0.804 | 0.701 | 0.498 | 0.588 | 0.574 | 0.638 | 0.707 | 0.618 |
| (2) | 0.531 | 0.562 | 0.426 | 0.952 | 0.804 | 0.720 | 0.481 | 0.602 | 0.535 | 0.667 | 0.726 | 0.636 |
| (3) | **0.556** | **0.562** | **0.565** | **0.963** | **0.840** | **0.720** | **0.498** | **0.608** | **0.604** | **0.667** | **0.735** | **0.664** |

## C  More Ablation Analysis

**The Efftiveness of EMMS under Various Measurements.** In addition to weighted Kendall's tau, we employ various other measures to evaluate our EMMS. These include Kendall's tau ($\tau$), Pearson's correlation ($r$), weighted Pearson's correlation ($r_w$), and top-$k$ relative accuracy, denoted as Rel@$k$, which represents the ratio between the best fine-tuning accuracy achieved on the downstream task using the top-k ranked models and the best fine-tuning precision achieved with all models. We test the robustness of our transferability metrics to different measurements on the Flickr8k and RSICD datasets for image captioning tasks, as shown in Table 17. Our EMMS consistently outperforms the previous transferability metric, including LogME and TransRate. Under the aforementioned measurements, demonstrating the superiority of our EMMS.

**The Effect of Label Embedding** In some multimodal tasks or text tasks, including image captioning or text question answering. Label emebdding directly affects the applicability of existing model selection metric to these tasks. In addition, even in classification tasks, the use of F-Label can also bring improvements in results. Here we focus on the comparison between label embedding and direct one-hot vectors for image classification tasks in CNN-based models. As shown in Table 18, the use of F-Label can bring performance improvement compared to One-Hot vector, the average $\tau_w$ increase from 0.618 to 0.636; furthermore, the use of multiple F-Label also brings some improvement compared to the average of single F-Label with $\tau_w$ increasing from 0.636 to 0.664.

**The Effect of Computational Speedup.** Here we experimentally demonstrate the effect of our accelerated algorithm. As shown in Table 21, the algorithm is similar to the in-accelerated version in terms of results, but much shorter in terms of the wall-clock time.

**Comparison with different number of iterations** The number of iterations affects the EMMS time, here we conduct experiments on the VQA task for the effect of the number of iterations on the results. As shown in Table .19, we find that the number of iterations does not have a large impact on the performance of our method, and even a small number of iterations can guarantee the final result(e.g. the number of iterations is 1). We believe that firstly our method converges very fast. And secondly, for the ranking problem of model ranking, even if the convergence is not sufficient, the original order can still be maintained to a large extent in EMMS, thus ensuring the effect.

Table 19: The effect of the number of iterations $r$ on VQA models in rank correlation $\tau_w$. We find that even a small number of iterations allows the method to maintain its effect.

|  | DAQUAR | COCO | CLEVR | DAQUAR | COCO | CLEVR |
|---|---|---|---|---|---|---|
|  | Weighted Kendall's tau $\tau_w$ | | | Wall-Clock Time (s) | | |
| $r = 3$ | **0.743** | 0.812 | 0.804 | 111.05 | 735.21 | 745.11 |
| $r = 2$ | 0.712 | 0.812 | 0.804 | 78.01 | 536.45 | 573.22 |
| $r = 1$ | 0.712 | **0.812** | **0.804** | **50.54** | **263.72** | **274.56** |

**The effect of using a single foundation model** We investigate how EMMS is influenced when only a single foundation model is provided. We conduct experiments on image classification and image captioning. We consider EMMS with the single foundation model including language foundation model (1) GPT-2 [27], (2) BERT [43] , (3) RoBerta [68], and multimodal foundation model (4) CLIP [26], (5) FLAVA [69], and (6) AltCLIP [70]. For comparison, we include the result of our EMMS with default setting (K=3, i.e. CLIP, BERT, and GPT-2) and the result of previous state-of-the-art methods obtained from LogME, NLEEP and TransRate. The results are reported in Table 20 and Table 5.

We have several observations. (1) Different downstream tasks prefer F-Labels obtained from different foundation models. No single foundation model is dominant in all target tasks. In particular, CLIP is not the best model for extracting F-Labels. (2) For image classification, both language and multimodal foundation models are competent for acquiring F-Labels. (3) Our EMMS can achieve the best results by combining F-Labels obtained from multiple foundation models.

Table 20: The effect of the single foundation model on EMMS. The results are obtained on image classification regarding $\tau_w$.

|  | Aircraft | Caltech | Cars | CF-10 | CF-100 | DTD | Flowers | Food | Pets | SUN | VOC | Avg. | SOTA/All |
|---|---|---|---|---|---|---|---|---|---|---|---|---|---|
|  | Weighted Kendall's tau $\tau_w$ | | | | | | | | | | | | |
| Previous SOTA | 0.299 | 0.412 | 0.693 | **0.741** | 0.736 | **0.621** | **0.655** | 0.580 | 0.707 | **0.619** | 0.651 | 0.610 | 4/11 |
| (1) Gpt2 | **0.481** | **0.463** | 0.448 | 0.652 | **0.745** | **0.621** | 0.562 | 0.652 | **0.740** | 0.616 | **0.730** | 0.610 | 6/11 |
| (2) Bert | **0.481** | 0.444 | 0.458 | 0.718 | **0.745** | **0.621** | 0.562 | 0.592 | **0.740** | 0.616 | **0.730** | 0.609 | 5/11 |
| (3) RoBerta | 0.448 | 0.444 | 0.507 | 0.701 | **0.745** | 0.608 | 0.562 | 0.580 | **0.740** | 0.574 | **0.730** | 0.604 | 3/11 |
| (4) CLIP | **0.481** | 0.444 | 0.496 | 0.608 | 0.720 | **0.621** | 0.562 | 0.558 | **0.740** | 0.616 | 0.706 | 0.595 | 3/11 |
| (5) FLAVA | **0.481** | 0.444 | 0.508 | **0.741** | **0.745** | **0.621** | 0.562 | 0.652 | **0.740** | 0.574 | 0.706 | 0.615 | 5/11 |
| (6) AltCLIP | **0.481** | 0.444 | 0.437 | **0.741** | **0.745** | **0.621** | 0.562 | 0.580 | **0.740** | 0.595 | **0.730** | 0.607 | 6/11 |
| EMMS | **0.481** | 0.444 | **0.706** | 0.718 | **0.745** | **0.621** | 0.562 | **0.673** | **0.740** | **0.619** | **0.730** | **0.639** | 8/11 |

**The Wall-clock Time of Label Embedding.** For classification tasks, since the maximum number of categories is often only a few hundred, Label Embedding is very fast. Here we focus on documenting the time required for multimodal tasks, e.g. image captioning, text question answering, and referring expression comprehension, where label embedding is more time-consuming. For each task, we use 8 Nvidia A100 GPUs for label embedding, with a batch size of 512 for each GPU. The running time of label embedding for image captioning, text question answering, and referring expression comprehension is shown in Table 22. We measure the time for each dataset on the same CPU device (AMD EPYC 7H12 with 64-Core Processor) for three times and take the average as the final result.

Table 21: The effect of computational speedup in image classification with ViT models. We can see that the accelerated version of the algorithm achieves a significant reduction in time while guaranteeing results. Two variants of EMMS are considered: (1) EMMS with normal algorithm; (2) EMMS with fast algorithm.

|  | Aircraft | Caltech | Cars | CF-10 | CF-100 | DTD | Flowers | Food | Pets | SUN | VOC | Avg. |
|---|---|---|---|---|---|---|---|---|---|---|---|---|
| | | | | | Weighted Kendall's tau $\tau_w$ | | | | | | | |
| (1) | **0.564** | **0.463** | 0.706 | 0.718 | 0.745 | 0.589 | **0.592** | 0.531 | **0.755** | 0.532 | 0.730 | 0.629 |
| (2) | 0.481 | 0.444 | **0.706** | **0.718** | **0.745** | **0.621** | 0.562 | **0.673** | 0.740 | **0.619** | **0.730** | **0.639** |
| | | | | | Wall-Clock Time (s) | | | | | | | |
| (1) | 102.06 | 114.72 | 177.25 | 718.34 | 724.5 | 50.24 | 87.28 | 944.57 | 83.37 | 336.92 | 104.9 | 313.10 |
| (2) | **21.31** | **17.23** | **28.06** | **154.61** | **182.11** | **13.87** | **15.95** | **265.99** | **17.93** | **63.86** | **16.63** | **72.55** |

Table 22: The wall-clock time (s) of label embedding in image captioning on 5 target tasks, text question answering on 2 target tasks, and referring expression comprehension on 3 target tasks,respectively.

| Task | | Image Captioning | | | | Text QA | | Referring EC | | |
|---|---|---|---|---|---|---|---|---|---|---|
| Dataset | F8k | F30k | RSD | F10k-H | F10k-R | SQuAD1.1 | SQuAD2.0 | RefCOCO | RefCOCO+ | RefCOCOg |
| Time | 14.56 | 89.31 | 18.92 | 3.37 | 3.13 | 35.67 | 53.87 | 49.19 | 48.88 | 31.63 |

**The computational complexity of EMMS.** We compare the computational complexity between LogME and EMMS in Table 23. We see that EMMS has lower computation complexity than LogME(F) because LogME(F) needs several iterations (T=3 on average) to converge. Moreover, EMMS allows for full vector computation and can be efficiently solved by existing scientific computation packages such as np.linalg.lstsq. Nevertheless, LogME(F) cannot be written in fully vectorized form because the model parameters in LogME(F) are highly coupled. Hence, LogME(F) can only be excuted in a while loop.

In addition, in the classification task, we compare EMMS and LogME. EMMS usually has higher computation complexity because $D_2 \gg C$. In some cases, when the number of categories $C$ and the iteration number $T$ are large, EMMS could be faster than LogME with vector computation. For example, we find that $C = 397$ and $T = 4.46$ on average over all models when LogME is convergent on the Sun397 dataset. It results in higher time complexity than LogME, as indicated in Table 23. We further verify this by implementing LogME with $T = 1$. As shown in Table 24, EMMS spends more time in calculating the transferability than LogME (T=1) on all datasets. However, LogME performs much worse than EMMS because it does not converge when $T = 1$.

Table 23: The comparison of computational complexity between LogME, EMMS(one), and EMMS in image classification. We denote model feature $X \in R^{N \times D_1}$ and F-labels $Z \in R^{N \times D_2 \times K}$ with $N \approx 10^4$, $D_1 \approx 10^3$, $D_2 = 1024$, $K = 3$, and $C \approx 10^2$. Moreover, $T \approx 3$ denotes the iteration number of LogME. Moreover, LogME(F) denotes LogME with F-Label.

| | Complexity | Simplified Complexity | Vector Compuration |
|---|---|---|---|
| LogME | $3TCD_1^2 + (2T+1)NCD_1 + D_1^3 + ND_1^2$ | $3TCD_1^2 + ND_1^2 + ND_1C(2T+1)$ | |
| LogME(F) | $3TD_2D_1^2 + (2T+1)NCD_1 + D_1^3 + ND_1^2$ | $3TD_2D_1^2 + ND_1^2 + ND_1D_2(2T+1)$ | |
| EMMS(One) | $CD_1^2 + NCD_1 + D_1^3 + ND_1^2$ | $ND_1^2$ | |
| EMMS | $ND_1^2 + 2ND_1D_2 + D_1^3 + D_1^2D_2 + (K^2+K)(ND_2) + K^3 + K^2 + K\log K$ | $ND_1^2 + 2ND_1D_2$ | |

Table 24: The comparison between LogME and EMMS. The results are obtained on image classification regarding $\tau_w$. LogME ($T = 1$) indicates that the inner loop of LogME only performs once.

| | Aircraft | Caltech | Cars | CF-10 | CF-100 | DTD | Flowers | Food | Pets | SUN | VOC |
|---|---|---|---|---|---|---|---|---|---|---|---|
| Weighted Kendall's tau $\tau_w$ | | | | | | | | | | | |
| LogME ($T = 1$) | 0.378 | 0.341 | -0.408 | 0.645 | 0.727 | 0.112 | -0.074 | 0.561 | 0.528 | 0.259 | -0.04 |
| LogME | 0.299 | 0.382 | 0.633 | **0.741** | 0.727 | 0.569 | 0.512 | 0.580 | 0.528 | 0.619 | 0.591 |
| EMMS(One) | 0.412 | 0.444 | 0.565 | 0.740 | 0.736 | **0.621** | **0.562** | 0.579 | **0.740** | 0.592 | **0.730** |
| EMMS | **0.481** | **0.444** | **0.706** | 0.718 | **0.745** | **0.621** | **0.562** | **0.673** | **0.740** | **0.619** | **0.730** |
| Wall-Clock Time (s) | | | | | | | | | | | |
| LogME ($T = 1$) | 4.45 | 4.72 | 8.18 | 34.81 | 40.15 | 3.65 | 5.13 | 53.7 | 4.59 | 31.66 | 6.03 |
| LogME | 8.93 | 10.89 | 30.28 | 53.07 | 62.13 | 4.78 | 9.27 | 104.92 | 6.28 | 425.43 | 7.42 |
| EMMS(One) | **4.12** | **4.45** | **8.07** | **19.45** | **26.18** | **2.65** | **4.03** | **39.72** | **3.50** | **24.84** | **4.07** |
| EMMS | 21.31 | 17.23 | 28.06 | 154.61 | 182.11 | 13.87 | 15.95 | 265.99 | 19.73 | 63.86 | 16.63 |

**Comparison with variants of existing methods.** To further validate the efficacy of EMMS, we compare it with TransRate using F-Labels on image classification. To this end, we estimate the mutual information of the model feature and F-label following TransRate. Specifically, denote model feature $X \in R^{N \times D_1}$, and the F-Label $Z_k \in R^{N \times D_2}$, we estimate the mutual information of $X$ and $Z_k$ after the discretization operation for each dimension of $D_2$ separately and then take average to obtain the final score.

Moreover, we implement two baselines based on TransRate. When $K = 1$, we instantiate the F-Label as the CLIP embedding. When $K = 3$, we instantiate the F-Labels as the embedding collection extracted from the CLIP, BERT, and GPT-2. In this case, the final score is averaged over three F-Labels. The results are shown in Table 25, where we can see that our EMMS consistently outperforms TransRate with F-Labels (both K=1 and K=3).

Table 25: The comparison between TransRate and EMMS. The results are obtained on image classification regarding $\tau_w$. TransRate ($K$) indicates that the number of foundation models used.

| | Aircraft | Caltech | Cars | CF-10 | CF-100 | DTD | Flowers | Food | Pets | SUN | VOC | Avg. |
|---|---|---|---|---|---|---|---|---|---|---|---|---|
| Weighted Kendall's tau $\tau_w$ | | | | | | | | | | | | |
| TransRate(K=1) | 0.297 | 0.440 | 0.682 | 0.655 | 0.501 | 0.533 | 0.548 | 0.537 | 0.736 | 0.533 | 0.666 | 0.557 |
| TransRate(K=3) | 0.295 | 0.441 | 0.682 | 0.523 | 0.501 | 0.542 | 0.548 | 0.539 | 0.730 | 0.533 | 0.679 | 0.546 |
| EMMS | **0.481** | **0.444** | **0.706** | **0.718** | **0.745** | **0.621** | **0.562** | **0.673** | **0.740** | **0.619** | **0.730** | **0.639** |

