*Proof.* Since we fix $w$ to optimize $t$ at this point, we define $s(t) = s(w, t)$, thus, $\nabla s(t) = -2Z^T(Xw^\star - Zt)$. For any $t_1, t_2 \in \text{dom}(s)$

$$\|\nabla s(t_1) - \nabla s(t_2)\| = \|2Z^T Z t_1 - 2Z^T Z t_2\| \le \|2Z^T Z\| \|t_1 - t_2\|.$$

According to the definition 1, it shows that the $f(t)$ is $\beta$-smooth, where $\beta = \|2Z^T Z\|$. We denote $t \in \triangle^{K-1}$ to be the initial point and $t^+$ to be the result of one iteration of $t$, where $t^+ = \Pi_{\triangle^{K-1}}(t - \frac{1}{\beta}\nabla f(t))$. From Lemma 3, we can replace $x^+, y$ and $x$ with $t^+, t$, and $t$, repsectively. In this way, the inequality holds

$$0 \le s(t^+) \le s(t) - \frac{1}{2\beta}\|\beta(t - t^+)\|^2 \le s(t)$$

85 $\qquad\qquad\qquad\qquad\qquad\qquad\qquad\qquad\qquad\qquad\qquad\qquad\qquad\qquad\qquad\qquad\qquad\quad\square$

86 Therefore, according to **Monotone convergence theorem**, the iterative optimization in the algorithm
87 for $t$ is convergent

88 **Theorem 2.** *Suppose* $s(w, t) = \frac{1}{2}\|Xw - Zt\|_2^2$ *where* $X \in \mathbb{R}^{N \times D}$, $Z \in \mathbb{R}^{N \times K}$, $w \in \mathbb{R}^{D \times 1}$ *and*
89 $t \in \triangle^{K-1}$, *the function value in Algorithm will be convergent. Specifically, denote* $w^\star, t^\star$ *as the*
90 *result after one iteration of* $w, t$ *respectively, we have* $0 \le s(w^\star, t^\star) \le s(w^\star, t) \le s(w, t)$.

91 *Proof.* In the first step, we denote $t \in \triangle^{K-1}$ is the initial point, then use gradient descent algorithm
92 to calculate $w^\star$. Since the optimization problem for $w$ is a convex optimization problem and use
93 lemma 2, the decreasing property for the gradient part can be derived. That is, for each $w \in \mathbb{R}^{D \times 1}$,
94 we have $s(w^\star, t) \le s(w, t)$. In the second step, we fix $w$ as $w^\star$, from Theorem 1, we have
95 $s(w^\star, t^\star) \le s(w^\star, t)$. Therefore, the value of $s(w, t)$ satisfies: $0 \le s(w^\star, t^\star) \le s(w^\star, t) \le s(w, t)$,
96 from **Monotone convergence theorem**, $s(w, t)$ converges to the limiting point. As shown above, the
97 overall convergence of our algorithm is guaranteed. $\qquad\qquad\qquad\qquad\qquad\qquad\qquad\square$

## C   Experiment

99 In this section, we present more experimental results in Sec. C.1, detailed descriptions of datasets
100 in Sec. C.2, pre-trained models and baselines in Sec. C.3, and ground-truth scores in Sec. C.4