# OpenReview forum: "Foundation Model is Efficient Multimodal Multitask Model Selector"
_NeurIPS.cc/2023/Conference — NeurIPS 2023 poster_

### Official Review · Reviewer_4U2K · 2023-07-06

**Soundness:** 2 fair
**Presentation:** 2 fair
**Contribution:** 3 good
**Rating:** 6
**Confidence:** 4

**Summary:**

This paper proposed an efficient multi-task model selector (EMMS) to address the inapplicability in a multi-modal multi-task scenario. Specifically, the proposed method achieves a new state-of-the-art in performance and speedup through the incorporation of design elements such as the F-label, Weighted Linear Square Regression, Fast Computation by Alternating Minimization.

**Strengths:**

1.	The motivation for addressing “a unified representation to represent diverse label formats” is clearly presented and validated. And use foundation model is an impressive way.
2.	The detailed derivation and experiment in this paper are comprehensive, providing strong evidence of the validity of the proposed model.

**Weaknesses:**

1.	It is recommended to include a complete demo in the code address of the paper, rather than a short python file.
2.	The phrase "5.13×, 6.29×, 3.59×, 6.19×, and 5.66× speedup in wall-clock time" appears to describe the speed efficiency of the EMMS (One) method. Could you clarify why EMMS's performance varies, being faster than LogME in some instances and slower in others, as indicated in Table 1?
3.	In Table 1, there seems to be some ambiguity about the effect of F-label on transferability assessment, since the perforamnce of the EMMS(one) and EMMS is identical in some rows.
4.	It is recommended that the paper include a broader array of foundational models for F-label. Is CLIP currently the model with the highest effects?

**Questions:**

na

---

> ### Author Rebuttal · Authors · 2023-08-09
>
> We would like to thank the reviewer for the useful and helpful comments on our manuscript.  We address the reviewer's concern below.
>
> **Q1:** "It is recommended to include a complete demo in the code."
>
> **A1:** Thanks for your suggestion. We have open-sourced complete code at https://github.com/anonymous123654/AnonymousEMMS.
>
> **Q2:** "The phrase "5.13×, ... in wall-clock time" appears to ... EMMS (One) method. Could you clarify why EMMS's performance varies, being faster than LogME in some instances and slower in others, as indicated in Table 1?"
>
> **A2:** It is true that we report the results (both $\tau_w$ and wall-clock time) of EMMS(one) in Table 1 for classification tasks. EMMS(one) is a variant of EMMS which replaces the F-Label with a one-hot label. Hence EMMS(one) can only be used in classification tasks. In image classification, we prefer to use EMMS(one) rather than EMMS because EMMS(one) achieves a good trade-off between effectiveness and efficiency. To see this, we list the computational complexity of LogME, EMMS(one), and EMMS in Table A.
>
> From Table A, we see that EMMS(one) have lower computational complexity than LogME. Moreover, our EMMS(one) allows for full vector computation and can be efficiently solved by existing scientific computation packages such as np.linalg.lstsq in our implementation. Nevertheless, LogME cannot be written in fully vectorized form because the optimization of model parameters of LogME are highly coupled.
>
> In addition, EMMS usually has higher computation complexity because $D_2 \gg C$. In some cases, when the number of categories $C$ and the iteration number $T$ are large, EMMS could be faster than LogME with vector computation. For example, we find that $C=397$ and $T=4.46$ on average over all models when LogME is convergent on the Sun397 dataset. It results in higher time complexity than LogME, as indicated in Table A. We further verify this by implementing LogME with $T=1$. As shown in Table B in the attached PDF, EMMS spends more time in calculating the transferability than LogME (T=1) on all datasets. However, LogME performs much worse than EMMS because it does not converge when $T=1$.
>
> **Table A.** The comparison of computational complexity between LogME, EMMS(one), and EMMS in image classification. We denote model feature $X \in R^{N \times D_1}$, F-labels $Z\in R^{N\times D_2\times K}$, and one-hot label $Y\in R^{N\times C}$ with $N \approx 10^4$, $D_1 \approx 10^3$, $D_2 \approx 10^3$, $K=3$, and $C\approx 10^2$. Moreover, $T$ denotes the iteration number of the inner loop in LogME with $T \approx 3$.
> | Method                             | LogME                             | EMMS（One）                       | EMMS                                                         |
> | --------------------------------- | --------------------------------- | --------------------------------- | ------------------------------------------------------------ |
> | **Complexity** | $3TCD_1^2 + (2T+1)NCD_1 + D_1^3 + ND_1^2$ | $CD_1^2 + NCD_1 + D_1^3 + ND_1^2$ | $ND_1^2 +2ND_1D_2 + D_1^3 + D_1^2D_2 + (K^2+K)(ND_2) + K^3 + K^2 +K\log K$ |
> | **Simplified Complexity** |  $3TCD_1^2 + ND_1^2+ND_1C(2T+1)$ | $ND_1^2$  | $ND_1^2 + 2ND_1D_2$  |
> | **Vector Compuration** |  ✗ | ✔  | ✔  |
>
>
> **Q3:** "Table 1, there seems to be some ambiguity about the effect of F-label on transferability assessment, since the performance of the EMMS(one) and EMMS is identical in some rows."
>
> **A3:** Thanks for the question. We claim that F-Label is the key ingredient of our EMMS. We demonstrate the effectiveness of F-Label in the following.
>
> F-Label has evident advantages over one-hot labels from our experiments.  First, from Table 1 of the main text, EMMS (with F-Label) outperforms EMMS(One) on 5 datasets but performs worse than EMMS(One) on only 1 dataset. Moreover, EMMS (with F-Label) has higher $\tau_w$ than EMMS(one) on average over 11 datasets in total. Second, we have compared the effectiveness of F-Label and one-hot label when selecting pre-trained CNN-based models in Table 10 of the Appendix. We see that EMMS (with F-Label) outperforms EMMS(one) on 7/11 datasets when a single F-Label is used and outperforms EMMS(one) on 9/11 datasets when multiple F-Labels are used. These results demonstrate the effectiveness of F-Labels.
>
> In addition, we also show that F-Label encodes richer semantic information than one-hot labels. Hence, EMMS has an advantage in encoding semantic labels with a large category size in image classification. For instance, EMMS outperforms EMMS(one), such as CF100 (EMMS 0.736 v.s. EMMS(one) 0.745), Food (EMMS 0.579 v.s. EMMS(one) 0.673) and Sun397 (EMMS 0.592 v.s. EMMS(one) 0.619) in Table 1 of the main text. Besides, F-Label unifies diverse forms of labels in different tasks and facilitates multitask model selection. As shown in Table 2-5 of the main text, EMMS is a good model selector in image captioning, VQA, TQA, and referring expression comprehension.
>
> **Q4:** "It is recommended that the paper include a broader array of foundational models for F-label. Is CLIP currently the model with the highest effects?"
>
> **A4:**  We have investigated the effect of different foundation models, including three language foundation models and three multimodal foundation models. Please see more details in [General Response Q1/A1](https://openreview.net/forum?id=2ep5PXEZiw&noteId=kK4DhbM069).
>
> We hope our responses help address the concerns of the reviewer. We are happy to run more experiments if the reviewer has any pieces of interest.

---

> ### Author Response · Authors · 2023-08-14
>
> Thank you very much for your insightful suggestions on this paper. We have responded to each of these in great detail. If you have any other questions, we are more than happy to provide additional clarification as well as experiments, and look forward to your reply !

---

> ### Author Response · Authors · 2023-08-17
>
> We express our sincere appreciation for the valuable suggestions regarding this paper. In response, we have provided thorough and detailed explanations. If there are any further questions, we are delighted to offer additional clarifications and conduct further experiments. We eagerly look forward to your reply!

---

> ### Author Response · Authors · 2023-08-19
>
> We would like to extend our heartfelt gratitude for the invaluable suggestions provided for this paper. In light of your feedback, we have diligently provided comprehensive and elaborate explanations. If you have any further inquiries, we are more than delighted to offer additional clarifications and conduct further experiments. We eagerly anticipate your response!

---

### Official Review · Reviewer_gNw5 · 2023-07-07

**Soundness:** 2 fair
**Presentation:** 2 fair
**Contribution:** 2 fair
**Rating:** 5
**Confidence:** 3

**Summary:**

This paper focus on an under-explored problem of estimating neural network transferring capability without actually fine-tuning the multi-modal multi-task model on individual downstream tasks. The problem is well-motivated and is of great practical importance. The solution proposed in this paper is straightforward, which essentially is to encode the target label (in the text form) with a foundation model. The embedding of the label text is then treated as the target and a model's transferability can be estimated through a simple weighted linear regression, which is then solved by an alternating minimization algorithm. The paper experiments with 5 downstream tasks and 24 datasets, and show that the proposed estimation method (EMMS) is fast and effective on most tasks.

**Strengths:**

1. The task is practically important and under-explored in previous work. As the authors mentioned, the previous estimation is usually limited by the classification tasks and the proposed work is more flexible as it directly encodes the label as a text sequence, therefore it can be used for multiple different types of tasks.
2. The proposed method is intuitive and straightforward, and the algorithms for solving the problem are clear. The theoretical proof looks reasonable although I didn't carefully check all the details in the equations.
3. The approximation with algorithm significantly speed up the computation when estimating the performance.
4. This paper performs extensive experimentations by fine-tuning the expensive models on many downstream tasks (to obtain ground truth) and the proposed approach shows superior performance over other methods with regard to correlation and the wall clock speed.

**Weaknesses:**

1. The Figure 1 bottom is not very informative -- the radar chart is most effective for showing multivariate data with quantitative variables, but in this setting the variables are all binary (applicable/inapplicable), therefore it's actually carrying limited information. Probably replace the figure with a table of checkbox to illustrate the advantage of the proposed approach.
2. As I mentioned in the limitations, some important assumptions are not validated in the paper. Specifically the key assumption that directly leads to the weighted linear square regression problem and alternating minimization algorithm is the linear mapping from model feature space to the F-label space. More justifications about the assumptions are needed.
3. I'm also concerned about the choice of evaluation metric of weighted Kendall's $\tau_w$. If I understand correctly, as a ranking metric it focus more on the relative orders of the elements in the rank, but ignores the actual number. In other words for A>B>C>D, it doesn't care whether A is just marginally higher than B or is much higher than B. It seems wasteful since we already obtain the ground truth downstream task model performance through expensive fine-tuning, but with the $\tau_w$ metric only the relative orders of the ground truths are useful. Therefore I'm wondering if it's possible to directly estimate the actual performance. I might be wrong or missed some important considerations so any clarifications or rationales are appreciated.
4. Some minor issues: in the tables with actual results, some numbers are bolded as the best performance, but in many cases there is a tie between two numbers in the table. Probably it's better to highlight both.

**Questions:**

I'm curious if there are other metrics that can be used to compare the proposed methods with other approaches, e.g. other than $\tau_w$ and wall-clock time. For the former as I stated in the weaknesses section, it seems that it only cares about relative orders of candidate pre-trained models; for the latter the wall-clock time is easily affected by multiple factors (e.g. multi-threading, concurrent execution of other tasks in multi-core environments, etc.) I would like to learn more about the rationale for choosing these metrics.
Also it seems in this paper only the linear mapping is considered and it is used as an assumption without further justification. This may need more clarifications as one can easily come up with alternative approaches for estimate the mapping from features to label embeddings.

**Limitations:**

This paper has a dedicated section to discuss the limitations and I agree with the authors that the quality of the foundation model (which is mainly responsible for the encoding of the label text) plays critical roles in the effectiveness. However I think there exists some other limitations which are also worth mentioning. Specifically, some important assumptions in this paper is in L164 and L168, where the label embedding is assumed as a linear mapping of the model feature. This assumption seems harsh and I think a more thorough discussion about the assumption will be useful. There is no obvious potential negative societal impact with this paper.

---

> ### Author Rebuttal · Authors · 2023-08-09
>
> We thank the reviewer for the detailed comments and valuable suggestions. We address the reviewer's concern as follows.
>
> **Q1:** "The Figure 1 bottom is not very informative ..."
>
> **A1:** Good advice. We have redrawn a table of checkboxes to illustrate the ability of EMMS. Please see Fig.A and Table A in the attached PDF for more details.
>
> **Q2:** "... need justification for the assumption of linear mapping ..."
>
> **A2:** We are sorry for the insufficient justifications. Here we provide more details about the linear assumption. Specifically, EMMS assumes that the true label embedding $z$ is a linear mapping of the model feature with Gaussian noise. This assumption is commonly used in recent methods. For example, LogME [1] assumes that: $z \leftarrow N(w^T\hat{x},\beta^{-1})$, which implies that there is a linear mapping from the model feature space to the label space. PACTran [2] also has a similar setting.
>
> The difference is that LogME takes a one-hot label as the true label embedding, which limits its applicability. But our EMMS treat the label embedding $z$ as a hidden variable. And F-Labels $\{z_k\}_{k=1}^K$ obtained from different foundation models are assumed to be noisy oracles of true label embedding $z$. Since labels in many tasks can be easily encoded into F-Lables, our EMMS can be used as a multitask model selector.
>
> The linear assumption is reasonable in image and text classification tasks because a linear classifier is usually used when the pre-trained model is transferred to a target task. For tasks of VQA and image captioning, the label in these tasks can be viewed as pure text [3]. By common practice [4,5], these tasks can also be essentially treated as a multi-label classification (vocabulary-based classification). Hence, a linear assumption is also reasonable for these tasks. For the task of referring to image comprehension, we still encode the bounding box label as embedding following [2]. We find that a linear assumption still works well, although the task head is a transformer encoder.
>
> It is noteworthy that the computation of previous model selection methods is computationally expensive if we direct employ them with vocabulary-based one-hot multiple labels due to the large vocabulary size. F-Labels in EMMS can be deemed as a compressed representation of vocabulary-based one-hot multiple labels as indicated in Fig.3. Due to the simplicity of a linear assumption and informative F-Labels, EMMS can quickly measure the transferability of numerous pre-trained models,  making it possible to be a fast and accurate transferability assessment.
>
> [1] Kaichao You, et al. LogME. In ICML 2021.
> [2] Nan Ding, et al. PACtran. In ECCV 2022.
> [3] Peng Wang, et al. OFA. In ICML 2022.
> [4] Antol et al. VQA. In ICCV 2015.
> [5] Goyal, Y., et al. Making the V in VQA matter. In: CVPR 2017.
>
> **Q3:** "I'm concerned about the choice of evaluation metric of weighted Kendall's ... if it's possible to estimate the actual performance."
>
> **A3:** Model selection is a well-defined problem in transfer learning, which aims to rank pre-trained models and select the top-performing model for the target task. Model selection techniques have been used extensively in classification tasks, such as LEEP, NLEEP, LogME, TransRate, PACTran, SFDA, and so on. The most common metric to measure the performance of a model selection method is the weighted Kendall's tau, i.e. $\tau_w$. Eqn. (15) of the Appendix gives the definition of $\tau_w$. In principle, a larger $\tau_w$ implies the transferability metric can rank pre-trained models better. And if a metric can rank top-performing models better, $\tau_w$ would also be larger. We also use other measurements, such as Person's coefficient, to assess the performance of transferability metrics in Table 9 of Sec. D in Appendix, where we can see that EMMS still outperforms LogME and other baselines under various measures.
>
> In addition, we agree that directly predicting the performance of a pre-trained model on the target task can be practical. But it is more challenging than model selection. Recent work [6] tackles this problem by proposing a benchmark consisting of ground-truth evaluations of 35 pre-trained models and 23 datasets. However, it still focuses on classification tasks, and it is unknown how well the prediction technique can be generalized to datasets in the training set. How to design a method to predict the performance on various target tasks requires ongoing efforts.
>
> [6] Orr Zohar et al. LOVM: Language-Only Vision Model Selection.
>
>
> **Q4:** "Some minor issues: in the tables with actual results, some numbers are bolded as the best performance, but in many cases, there is a tie between two numbers in the table. Probably it's better to highlight both."
>
> **A4:** Thank you for the suggestion. We will highlight all the best results in the final version.
>
>
> **Q5:** "For the latter, the wall-clock time is easily affected by multiple factors (e.g. multi-threading, concurrent execution of other tasks in multi-core environments, etc."
>
> **A5:** For the computation complexity, we measure the time for each metric on the same CPU device (AMD EPYC 7H12 with 64-Core Processor) three times and take the average as the final result. To avoid the influence of multi-threading, concurrent execution. We run each model selection method on a single dataset for a single testing case. The code for counting wall-clock time is available at https://github.com/anonymous123654/AnonymousEMMS. In addition, we also compare the theoretical computational complexity between EMMS and LogME in Table C of General Response Q2/A2. We can see that our EMMS not only has lower computation complexity but also enables fully vectorized computation. Please check more details in [General Response](https://openreview.net/forum?id=2ep5PXEZiw&noteId=kK4DhbM069).
>
> We hope our responses help address the concerns of the reviewer. We are happy to run more experiments if the reviewer has any pieces of interest.

---

> ### Author Response · Authors · 2023-08-14
>
> Thank you very much for your constructive suggestions about the paper, it definitely helped us to improve it. If you have any other questions we are more than willing to continue with the clarifications and experiments, looking forward to your reply!

---

> > ### Comment · Reviewer_gNw5 · 2023-08-16
> > **Thanks for the explanations!**
> >
> > I've read the response from the authors and I'm satisfied with the answers. I am gonna increase my rating for this paper.

---

### Official Review · Reviewer_z1Vu · 2023-07-10

**Soundness:** 3 good
**Presentation:** 3 good
**Contribution:** 3 good
**Rating:** 4
**Confidence:** 3

**Summary:**



This paper proposes to utilize large-scale foundation models for efficient multi-task model selector (EMMS). Concretely, the authors utilize foundation model to transform different label format (category label, text, bounding boxes) into unified noisy label embeddings. EMMS then could measure the compatibility between the models’ features and corresponding
59 label embeddings via weighted linear regression. Experiments on 5 downstream tasks with 24 multi-modal tasks shows the proposed method's effectiveness and efficiency.


**Strengths:**

- the motivation of this paper is important: investigating how to evaluate foundation model on a set of multi-modal tasks without finetuning all the target tasks.

- the writing is clear and easy to follow.

- the authors also provide code for reproducibility.

- The experiments show improvement on the 5 downstream tasks with 24 datasets.

**Weaknesses:**

As the authors mentioned in the limitation section, the proposed method is bottlenecked by the capabilities of the chosen foundation model. One could further ask, if the foundation model is good enough, why don't we just use the foundation model for the intended downstream tasks? For example, use CLIP for image classification?




**Questions:**

please refer to the weakness section

**Limitations:**

yes

---

> ### Author Rebuttal · Authors · 2023-08-09
>
> We thank the reviewer for the constructive feedback and helpful suggestions. We have provided a detailed general response to the concerns of all the reviewers. We address the reviewer's concern as follows.
>
> **Q1:** "As the authors mentioned in the limitation section, the proposed method is bottlenecked by the capabilities of the chosen foundation model. One could further ask, if the foundation model is good enough, why don't we just use the foundation model for the intended downstream tasks? For example, use CLIP for image classification?"
>
> **A1:** It is a significant question. It is known that foundation models have achieved great success in various tasks. Why do we still need to perform model selection on some conventional models? We have provided some explanations in related work in Sec. 2. Here, we answer this question from three aspects.
>
> **First, although foundation models have strong generalization ability, it still obtains suboptimal performance on some tasks.** For example, the original paper on CLIP (see Fig. 5) [1] points out that zero-shot CLIP is worse than ResNet-50 on several specialized, complex tasks such as satellite image classification (EuroSAT and RESISC45) and lymph node tumour detection (PatchCamelyon). On the other hand, many existing moderate-size pre-trained models can do some domain-specific tasks very well. Our EMMS is complementary to foundation models, serving as a tool to select an appropriate pre-trained model for target tasks.
>
> **Second, foundation models have amounts of parameters which are computationally expensive and require significant computational resources to train and deploy.** In certain scenarios, such as on mobile devices where computational resources are limited, conventional models that are smaller and faster to train can be more practical and cost-effective.
>
> **Third, model selection is a well-defined challenging task in transfer learning [2,3,4,5], considering that different tasks may have specific requirements.** However, previous approaches [2,3,4,5] mainly focus on classification tasks. In this work, we extend the model selection to multimodal multitask scenarios by label embedding encoded from multiple foundation models, which essentially expands the applicability of the model selection technique. Similar to TaskMatrix.AI [6], we believe that EMMS could be a useful tool for task completion with the help of foundation models.
>
> We hope these experiments help address the concerns of the reviewer. We are happy to run more experiments if the reviewer has any pieces of interest.
>
> [1] Alec Radford, et al. Learning Transferable Visual Models From Natural Language Supervision. In ICML 2021.
> [2] Kaichao You, et al. LogME: Practical assessment of pre-trained models for transfer learning. In ICML 2021.
> [3] Wenqi Shao, et al. Not all models are equal: Predicting model transferability in a self-challenging
> 375 fisher space. In ECCV 2022.
> [4] Yandong Li, et al. Ranking neural checkpoints. In CVPR 2021.
> [5] Nan Ding, et al. PACtran: PAC Bayesian Metrics for Estimating the Transferability of Pre-trained Models to Classification Tasks. In ECCV 2022.
> [6] Yaobo Liang et al. TaskMatrix.AI: Completing Tasks by Connecting Foundation Models with Millions of APIs.

---

> ### Author Response · Authors · 2023-08-14
>
> We appreciate the appreciation given by the reviewer towards our work. We are delighted to conduct additional experiments if the reviewer has any specific areas of interest. We eagerly await your response.

---

> ### Author Response · Authors · 2023-08-17
>
> We are grateful for the kind appreciation expressed by the reviewer regarding our work. To further enhance our research, we are more than willing to conduct additional experiments if you need. We eagerly look forward to hearing from you!

---

> ### Author Response · Authors · 2023-08-19
>
> We are thankful for the reviewer's acknowledgement of our efforts. We trust that the provided responses address the reviewer’s concerns regarding EMMS. We have presented sufficient explanations to clarify why we still need to perform model selection even at the time of the foundation model. We eagerly await any forthcoming questions and will be delighted to offer further clarification during the discussion stage.

---

### Official Review · Reviewer_vJKA · 2023-07-11

**Soundness:** 3 good
**Presentation:** 2 fair
**Contribution:** 3 good
**Rating:** 4
**Confidence:** 3

**Summary:**

This paper introduces EMMS, an efficient multi-task model selector for predicting the performance of pre-trained neural networks on multi-modal tasks without fine-tuning. EMMS employs large-scale foundation models to transform diverse label formats into a unified noisy label embedding. Through weighted linear regression and an alternating minimization algorithm, EMMS accurately estimates transferability. Experimental results demonstrate superior performance and significant speedup compared to existing methods.

**Strengths:**

1.	Generic Transferability Estimation Technique: The proposed method, Efficient Multi-task Model Selector (EMMS), offers a generic approach to estimate transferability. By utilizing a unified label embedding derived from foundation models and employing a simple weighted linear square regression (WLSR), EMMS becomes a fast and effective method for assessing the transferability of pre-trained models across different tasks.

2.	Novel Alternating Minimization Algorithm: The paper introduces a novel alternating minimization algorithm specifically designed to solve the WLSR problem. This algorithm ensures efficient and accurate estimation of transferability within the EMMS framework.

3.	The authors did extensive experiments across different tasks and the results demonstrate the efficacy and effectiveness of the proposed method.



**Weaknesses:**

1.	The proposed method is complicated and difficult to follow. I can barely understand how to use WLSR to maximize the log-likelihood, which may bring extra complexity for re-producing the method.
2.	One of the simple baselines could be estimating the mutual information between the $\hat{x}$ and $Z$. I doubt the effectiveness of this baseline, but it could be nice to have.
3.	More experiment details need to be provided like, what are the foundation models used in different tasks. Another ablation could be studying the effectiveness of different foundation models. I guess CLIP could be the strongest model to use in those tasks if we just use one single foundation model.

Minor:
Line 111, `denote finetuning score`.
Please provide more description of Figure 3 (b) as it is not straightforward to understand the confusion matrix of image captions.


**Questions:**

Please address the comments in weakness section.

---

> ### Author Rebuttal · Authors · 2023-08-09
>
> We thank the reviewer for the valuable suggestions. We list responses to the reviewer's concerns below.
>
> **Q1:** "The proposed method is difficult to follow..."
>
> **A1:** We are sorry for the vague presentation of our method. In principle, our EMMS is well established by maximizing the log-likelihood of regression with multiple noisy oracles. We describe the detailed setup as follows.
>
> **First**, EMMS assumes that the true label embedding $z$ is a linear mapping of the model feature with Gaussian noise. Moreover, F-Labels $z_k$ obtained from different foundation models are noisy oracles of $z$.  **Second**, by this setup, we write down the log-likelihood $\mathcal{L} = \sum_{(\hat{x},z_k)}\log P(z_1,\cdots, z_K|\hat{x})$ as given by Eqn. (2) of the main text.  **Third**, with further simplification of Eqn. (2), we find that maximizing the log-likelihood can be approximated as a WLSR problem, i.e. Eqn. (3) of the main text. We sincerely suggest the reviewer see the detailed derivations in Sec. B.1 of the Appendix.
>
> In addition, we claim that our EMMS is easy to implement and can be used in various multimodal tasks. Despite the complexity of derivations, EMMS turns out to tackle a simple WLSR problem, which can be solved with several lines of code by our algorithm, i.e. Algorithm 2 of the main text. For reference, we make our full code public at https://github.com/anonymous123654/AnonymousEMMS.
>
> **Q2:** "One of the simple baselines could be estimating the mutual information ...."
>
> **A2:** Thanks for the suggestion. We agree that it is important to compare our EMMS with the approach based on mutual information modelling. In fact, we have compared our EMMS with TransRate [1] in Table 1 of the main text, which models the relationship between the model feature and the one-hot label by mutual information. From Table 1, we can see that EMMS outperforms TransRate on almost all classification datasets.
>
> To further validate the efficacy of EMMS, we compare it with TransRate using F-Labels on image classification. We estimate the mutual information of the model feature and F-label following TransRate. Specifically, denote model feature $X \in R^{N \times D_1}$, and the F-Label $Z_k \in R^{N \times D_2}$, we estimate the mutual information of $X$ and $Z_k$ after the discretization operation for each dimension of $D_2$ and then take average to obtain the final score.
>
> Moreover, we implement two baselines based on TransRate. When $K=1$, we instantiate the F-Label as the CLIP embedding. When $K=3$, we instantiate the F-Labels as the embedding collection extracted from the CLIP, BERT, and GPT-2. In this case, the final score is averaged over three F-Labels. The results are shown in Table A, where we can see that our EMMS consistently outperforms TransRate with F-Labels (both K=1 and K=3).
>
> **Table A.** Comparison of different transferability metrics on image classification regarding $\tau_w$
> | Method       | Aircraft      | Caltech    | Cars      | CF10      | CF100     | DTD       | Flowers   | Food      | Pets      | SUN    | VOC   | Average   | Sota-All   |
> | ------------ | --------- | --------- | --------- | --------- | --------- | --------- | --------- | --------- | --------- | --------- | --------- |--------- |--------- |
> | TransRate(K=1) | 0.297     | 0.440     | 0.682     | 0.655     | 0.501     | 0.533     | 0.548     | 0.537     | 0.736     | 0.533     | 0.666     | 0.557     | 0-11     |
> | TransRate(K=3) | 0.295     | 0.441     | 0.682     | 0.523     | 0.501     | 0.542     | 0.548     | 0.539     | 0.730     | 0.533     | 0.679     | 0.546     | 0-11     |
> | **EMMS**     | **0.481** | **0.444** | **0.706** | **0.718** | **0.745** | **0.620** | **0.562** | **0.673** | **0.740** | **0.619** | **0.730** | **0.639**     | **11-11**     |
>
> [1] Long-Kai Huan, et al. TransRate. In ICML 2022.
>
> **Q3:** "More experiment details need to be provided like, what are the foundation models used in different tasks."
>
> **A3:** Thanks for the suggestion. We have provided the details about the foundation models used in different tasks in Sec. C of the Appendix. For reference, we present it in Table B. Note that CLIP is ineffective in extracting meaningful embedding for pure text tasks without a dedicated design [1]. We use ELECTRA [2] for text question answering instead of CLIP. Electra is a different type of language model which pre-trains a generator model and shows promising results in various NLP tasks.
>
> [1] Libo Qin et al. CLIPTEXT. In ACL 2023.
> [2] Kevin Clark et al. Electra. ICLR 2020.
>
> **Table B.** Foundation models used in different tasks.
> | Task   | Foundation Models  |
> | -------- | --------- |
> |ImgCls, ImgCap, Referring Expression Comprehension, VQA | CLIP, BERT, GPT-2|
> | Text QA | GPT-2, BART, ELECTRA |
>
>
> **Q4:** "Another ablation ... I guess CLIP could be the strongest model ... single foundation model."
>
> **A4:** We have investigated the effect of different foundation models, including three language foundation models and three multimodal foundation models. Please see more details in [General Response Q1/A1](https://openreview.net/forum?id=2ep5PXEZiw&noteId=kK4DhbM069).
>
>
> **Q5:** "More description of Figure 3 (b)."
>
> **A5:** We are sorry for the insufficient explanations. Fig3(b) shows the correlation between captions from different images. Each image has two captions denoted by the brace. Two captions from the same image often exhibit similarity. Note that the one-hot label for a caption is translated from a vocabulary following [3]. The corresponding entry is 1 if the word corresponds to one index of vocabulary. We see that it is easier for EMMS to encode the semantic relevance of two captions corresponding to the same image than one-hot encoding. We will incorporate it into the final version.
>
> [3] Nan Ding et al. PACtran. In ECCV 2022.
>
> We hope these experiments help address the concerns of the reviewer. We are happy to run more experiments if the reviewer has any pieces of interest.

---

> ### Author Response · Authors · 2023-08-14
>
> Thank you very much for your constructive suggestions. We have carefully considered and experimented with them. If you have any further questions, we are interested in conducting additional experiments on this topic. We look forward to hearing from you!

---

> ### Author Response · Authors · 2023-08-17
>
> We greatly appreciate the insightful suggestions provided by you. We have carefully taken them into account and conducted experiments accordingly. In case there are any further queries, we are enthusiastic about executing more experiments. We eagerly await your response!

---

> ### Author Response · Authors · 2023-08-19
>
> We greatly appreciate your helpful suggestions. We trust that the provided responses address the reviewer’s concerns regarding EMMS. We have presented more details about the mechanism behind EMMS and released the full code. We have conducted more experiments including a comparison with methods based on mutual information, and an ablation study of how each single foundation affects our EMMS. We eagerly await any forthcoming questions and will be delighted to offer further clarification during the discussion stage.

---

### Official Review · Reviewer_kTYc · 2023-07-18

**Soundness:** 3 good
**Presentation:** 3 good
**Contribution:** 3 good
**Rating:** 5
**Confidence:** 2

**Summary:**

This paper introduces the Efficient Multi-task Model Selector, which utilizes foundation models to convert diverse labels into unified label embeddings. These embeddings are then used to calculate a transferability metric within a weighted linear square regression (WLSR) framework. The proposed method achieves impressive results in multi-task scenarios.

**Strengths:**

1. The paper presents a novel and unified strategy for model selection. The use of label embeddings enables the capturing of fine-grained semantics, leading to improved estimation in downstream tasks.

2. The effectiveness of the proposed method is supported by a comprehensive set of experiments covering various downstream tasks.

3. The derivation of equations is provided, establishing a solid foundation for the main contribution of the paper.

**Weaknesses:**

No questions.

**Questions:**

No questions.

**Limitations:**

No questions.

---

> ### Author Rebuttal · Authors · 2023-08-09
>
> We thank the reviewer for the recognition of our work. We are happy to run more experiments if the reviewer has any pieces of interest.

---

> > ### Comment · Reviewer_kTYc · 2023-08-21
> >
> > Dear authors,
> >
> > Thanks for your response. I have thoroughly reviewed both your rebuttal and the feedback provided by the other reviewers. I acknowledge that the problem you've proposed is indeed interesting and holds value. However, it appears that the current version of the submitted paper lacks certain essential ablation experiments. Considering these factors, I have chosen to cast my vote as borderline accept, and I am eagerly anticipating the revised version of the paper.

---

> > > ### Author Response · Authors · 2023-08-21
> > > **Thank you for your reply**
> > >
> > > Thank you for your response. During the rebuttal, we conducted additional ablation experiments including (1) using more single foundation models to test the effect of EMMS, (2) additional baseline measured with mutual information and comparing the effect, and (3) comparing the computational complexity between LogME and EMMS.
> > >
> > > **Firstly**, for (1), we discover that different foundation models are preferred for different tasks and CLIP is not always the best foundation model. This is described in detail in [General Response A1](https://openreview.net/forum?id=2ep5PXEZiw&noteId=kK4DhbM069). **Secondly**, we add a new baseline in terms of mutual information metrics to illustrate the superiority of EMMS, which is displayed in [Reply to Reviewer vJKA A2](https://openreview.net/forum?id=2ep5PXEZiw&noteId=OKtcHuyCuX). **Lastly**, for (3),  we have carefully compared the time complexity of EMMS and LogME, as detailed in [Reply to Reviewer 4U2k A2](https://openreview.net/forum?id=2ep5PXEZiw&noteId=rxHsFLXxmy).
> > >
> > > Besides, the NeruIPS policy recounts that the rebuttal phase is a vital part of the review process, as it offers authors an opportunity to address concerns and clarify misunderstandings.  We think that the experiments added during the rebuttal should also be taken into account. We sincerely suggest the reviewer check our detailed response in the corresponding part.
> > >
> > >  Thanks for your suggestion again!

---

### Author Rebuttal · Authors · 2023-08-09

# General Response
We thank the reviewers for their detailed reviews and thoughtful suggestions on our work.

In general, two main concerns are raised, including (1) the effect of using a single foundation model and (2) the computational complexity of EMMS. To address the reviewers’ concerns, we try our best to perform additional experiments.

**Q1: The effect of using a single foundation model to extract F-Label. Is CLIP the best model?**

**A1:** We investigate the effect of using a single foundation model. We conduct experiments on image classification and image captioning. We consider EMMS with the single foundation model including language foundation model (1) GPT-2, (2) BERT, (3) RoBerta, and multimodal foundation model (4) CLIP, (5) FLAVA, and (6) AltCLIP. For comparison, we include the result of our EMMS with default setting (K=3, i.e. CLIP, BERT, and GPT-2) and the result of previous state-of-the-art methods obtained from LogME, NLEEP and TransRate. The results are reported in Table A and Table B.

We have several observations. (1) Different downstream tasks prefer  F-Labels obtained from different foundation models. No single foundation model is dominant in all target tasks. In particular, CLIP is not the best model for extracting F-Label. (2) For image classification, both language and multimodal foundation models are competent for acquiring F-Labels. For image captioning, multimodal foundation models are more appropriate for extracting F-Labels than language foundation models. (3) Our EMMS can achieve the best results by combining F-Labels obtained from multiple foundation models.



**Table A**: The effect of the single foundation model in image classification.
| Method   | Aircraft | Caltech | Cars      | CF10      | CF100     | DTD   | Flowers | Food      | Pets  | Sun397    | Voc2007   | Average   | Sota/All   |
| -------- | -------- | ------ | --------- | --------- | --------- | ----- | ------- | --------- | ----- | --------- | --------- | --------- | --------- |
| Previous SOTA | 0.299    | 0.412  | 0.693     | **0.741**     | 0.736     | **0.621** | **0.655**   | 0.580     | 0.707 | **0.619**     | 0.651     | 0.610    | 4/11     |
| (1) GPT2     | **0.481**    | **0.463**  | 0.448     | 0.652     | **0.745**     | **0.621** | 0.562   | 0.652     | **0.740** | 0.616     | **0.730**     | 0.610     | 6/11     |
| (2) Bert     | **0.481**    | 0.444  | 0.458     | 0.718     | **0.745**     | **0.621** | 0.562   | 0.592     | **0.740** | 0.616     | **0.730**     | 0.609     | 5/11     |
| (3) RoBerta  | 0.448    | 0.444  | 0.507     | 0.701     | **0.745**     | 0.608 | 0.562   | 0.580     | **0.740** | 0.574     | **0.730**     | 0.604     | 3/11     |
| (4) CLIP     | **0.481**    | 0.444  | 0.496     | 0.608     | 0.720     | **0.621** | 0.562   | 0.558     | **0.740** | 0.616     | 0.706     |  0.595     | 3/11     |
|(5) FLAVA     | **0.481** | 0.444 | 0.508     | **0.741** | **0.745** | **0.621** | 0.562 | 0.652     | **0.740** | 0.574     | 0.706     |  0.615    |  5/11    |
|(6) AltCLIP   | **0.481** | 0.444 | 0.437     | **0.741** | **0.745** | **0.621** | 0.562 | 0.580     | **0.740** | 0.595 | **0.730** |  0.607    |  6/11    |
| **EMMS (ours)**     | **0.481**    | 0.444  | **0.706** | 0.718 | **0.745** | **0.621** | 0.562   | **0.673** | **0.740** | **0.619** | **0.730** | **0.639** | 8/11 |

**Table B.** The effect of the single foundation model on EMMS in image captioning.
| Method        | Flickr8k  | Flickr30k | RSICD | flickr10kH | flickr10kR | Average   | Sota/All   |
| ------------- | --------- | --------- | ----- | ---------- | ---------- | ---------- | ---------- |
| LogME（CLIP） | 0.530     | 0.393     | 0.618 | 0.764    | 0.634      | 0.588    |   0/5    |
| (1) Gpt2          | 0.566     | 0.393     | 0.431 | 0.715      | 0.618      | 0.545      |  0/5     |
| (2) Bert          | 0.395     | 0.319     | 0.448 | **0.802**      | **0.711**      | 0.535      |  2/5    |
| (3) Roberta  | 0.346     | 0.111     | 0.587 | 0.571      | 0.566      | 0.436      | 0/5     |
| (4) CLIP      | 0.510     | 0.448     | **0.704** | **0.802**      | 0.678      | 0.628      | 2/5      |
| (5) FLAVA   | 0.463  | 0.382   | 0.693 | 0.704   | 0.678   | 0.584   | 0/5 |
| (6) AltCLIP    | 0.453     | 0.448     | 0.623     | **0.802**      | 0.678      | 0.601   | 1/5 |
| EMMS          | **0.660** | **0.504** | **0.704** | **0.802**      | 0.678      | **0.670**      | **4/5**|

**Q2: The computational complexity of EMMS.**

**A2:** We compare the computational complexity between LogME and EMMS in Table C. We see that EMMS has lower computation complexity than LogME because LogME needs several iterations (T=3 on average) to converge. Moreover, EMMS allows for full vector computation and can be quickly solved by existing packages such as numpy.linalg.lstsq. But LogME cannot be written in fully vectorized form as the model parameters in LogME are highly coupled. Hence, LogME needs to be executed in a while loop.

**Table C.** The comparison of computational complexity between LogME and EMMS. LogME is fed with F-Label. We denote model feature $X \in R^{N \times D_1}$ and F-labels $Z\in R^{N\times D_2\times K}$ with $N \approx 10^4$, $D_1 \approx 10^3$, $D_2=1024$, $K=3$, and $C\approx 10^2$. Moreover, $T\approx 3$ denotes the iteration number of LogME.
| Method                             | LogME                             |               EMMS                                                         |
| --------------------------------- | --------------------------------- | ------------------------------------------------------------ |
| **Complexity** | $3TD_1^2D_2+(2T+1)ND_1D_2+D_1^3+ND_1^2$  | $ND_1^2+2ND_1D_2+D_1^3+D_1^2D_2+(K^2+K)(ND_2)+K^3+K^2+K\log K$ |
| **Simplified Complexity** |  $3TD_1^2D_2+ND_1^2+ND_1D_2(2T+1)$  | $ND_1^2+2ND_1D_2$  |
| **Vector Compuration** |  ✗   | ✔  |

We are happy to run more experiments if the reviewer has any pieces of interest.

---

### Author Response · Authors · 2023-08-14

We would like to express our sincerest appreciation to the reviewers for their constructive comments and suggestions, which essentially strengthen our paper. We are pleased that the reviewers agree on the importance and novelty of the research problem we have addressed, the directness of our solution algorithm, extensive experiments, and solid theoretical analysis.

In general, the reviewers expressed some concerns about our paper, which can be summarized as follows. (1) The effectiveness of using a single foundation model and whether CLIP is the best foundation model. (2) The rationality of certain assumptions made in the paper, such as the linearity assumption. (3) The time complexity of the proposed method and the reasons for its variability compared to LogME. (4) More experiments to demonstrate the effectiveness of our EMMS.

To address these concerns, we have conducted additional experiments. Firstly, for (1), we have used more single foundation models to test the effectiveness of EMMS and discovered that CLIP is not always the best foundation model. This is described in detail in [General Response A1](https://openreview.net/forum?id=2ep5PXEZiw&noteId=kK4DhbM069). Secondly, for (2), we have pointed out the common use of the linear assumption in model selection problems, which is explained in [Reply to Reviewer gNw5 A2](https://openreview.net/forum?id=2ep5PXEZiw&noteId=Deq7ncXFXZ). Thirdly, for (3), we have carefully compared the time complexity of EMMS and LogME and provide a reasonable explanation for the observed variability, as detailed in [Reply to Reviewer 4U2k A2](https://openreview.net/forum?id=2ep5PXEZiw&noteId=rxHsFLXxmy). Lastly, for (4), we add a new baseline in terms of mutual information metrics to illustrate the superiority of EMMS, which is displayed in [Reply to Reviewer vJKA A2](https://openreview.net/forum?id=2ep5PXEZiw&noteId=OKtcHuyCuX).

We hope that our responses address the concerns of the reviewers. We are open to conducting more experiments if there are any specific areas of interest raised by the reviewer. We look forward to the reviewers' reply!

---

### Author Response · Authors · 2023-08-21
**Did not receive responses from Reviewer vJKA, Reviewer z1Vu, and Reviewer 4U2K so far**

Dear ACs,

We thank all reviewers for their valuable comments, which have substantially improved our work. We highlight **the contributions of EMMS** as follows.

- Our work tackles an important problem of model selection in transfer learning. The proposed EMMS is the first transferability estimating method which can be used in multimodal multitask scenarios.
- Equipped with a unified label embedding provided by foundation models, we show that multitask model selection can be simplified into a problem of weighted least square regression (WLSR). We propose a novel alternating minimization algorithm to solve WLSR
efficiently with theoretical analysis.
- Extensive experiments on 5 downstream tasks with 24 datasets demonstrate the effectiveness of EMMS. Specifically, EMMS achieves 9.0%, 26.3%, 20.1%, 54.8%, and 12.2%, performance gain on image recognition, referring, captioning, visual question answering, and text question answering, while bringing 5.13×, 6.29×, 3.59×, 6.19×, and 5.66× speedup in wall-clock time compared with the state-of-the-art method LogME.

Reviewer vJKA, Reviewer z1Vu, and Reviewer 4U2K raised some concerns about our manuscript, to which we have added sufficient experiments as well as detailed explanations during rebuttal. Unfortunately, we have not received any feedback from Reviewer vJKA, Reviewer z1Vu, and Reviewer 4U2K so far.

**Reviewer vJKA's** suggestions include: (1) conducting robustness analysis experiments for the effectiveness of the single foundation model. (2) adding a baseline that measures the performance using mutual information, and (3) providing a clear description of the WLSR process.  In response to these suggestions, we (1) provide detailed explanations of our method EMMS, (2) compare EMMS with TransRate which is a transferability estimating method based on mutual information, and (3) investigate the effect of each single foundation on EMMS.

**Reviewer z1Vu** raised a crucial question regarding why we don't directly use the foundation model for all tasks instead of performing the model selection. We addressed this concern by noting that EMMS could be a useful tool for task completion as foundation models still achieve suboptimal performance on certain tasks. Moreover, foundation models are computationally expensive and require significant computational resources for training and deployment.

**Reviewer 4U2K** requested (1) further experiments that involve a broader range of foundational models for F-Labels, echoing the issue raised by Reviewer vJKA, (2) the assessment of computational complexity of EMMS, and (3) improvements to the open-source code. In response, we (1) explain the time complexity differences between EMMS and LogME and highlighted the importance of F-Label for enabling the scalability of EMMS to multiple tasks, (2) conduct more experiments to demonstrate the enhancing effect of F-Label on EMMS, and (3) release the open-source code.

We believe that these explanations and additional experiments can answer their questions well. Hope our clarification helps you make a decision about our work.

Best Regards,

NeurIPS 2023 Conference Paper5953 Authors

---

### Decision · Program_Chairs · 2023-09-21

**Decision:**

Accept (poster)

**Comment:**

The paper addresses the challenge of predicting the performance of pre-trained neural networks on various multi-modal tasks without the need for fine-tuning. The authors present the Efficient Multi-task Model Selector (EMMS) which employs foundation models to unify diverse label formats into a cohesive label embedding. Main concerns are the complexity of the method and the reliance on foundation model capabilities. After discussion, main concerns are addressed. Given the general quality of the paper and the feedback, I recommend acceptance.